

# Analytical solution for residual stress and strain preserved in anisotropic inclusion entrapped in isotropic host

*Xin Zhong[1,2], Marcin Dabrowski[1,3], Bjørn Jamtveit[2]*

1. Institut für Geologische Wissenschaften, Freie Universität Berlin, Malteserstrasse 74–100, 12449 Berlin, Germany

2. Physics of Geological Processes, The Njord Center, University of Oslo, Norway

3. Computational Geology Laboratory, Polish Geological Institute - NRI, Wrocław, Poland

Correspondence to: Xin Zhong (xinzhong0708@gmail.com)

**Abstract.** Raman elastic thermobarometry has recently been applied in many petrological studies to recover the pressure-temperature (*P-T*) conditions of mineral inclusion entrapment. Existing modelling methods in petrology either adopt an
assumption of a spherical, isotropic inclusion embedded in an isotropic, infinite host, or use numerical techniques such as finite element method to simulate the residual stress and strain state preserved in the non-spherical anisotropic inclusion. Here, we use the Eshelby solution to develop an analytical framework for calculating the residual stress and strain state of an elastically anisotropic, ellipsoidal inclusion in an infinite, isotropic host. The analytical solution is applicable to any class of inclusion symmetry and an arbitrary inclusion aspect ratio. Explicit expressions are derived for some symmetry classes
including e.g. tetragonal, hexagonal and trigonal.

The effect of changing the aspect ratio on residual stress is investigated including quartz, zircon, rutile, apatite and diamond inclusions in garnet host. Quartz is demonstrated to be the least affected, while rutile is the most affected. For prolate quartz inclusion (*c*-axis longer than *a*-axis), the effect of varying the aspect ratio on Raman shift is demonstrated to be insignificant. When *c/a*=5, only ca. 0.3 cm[-1] wavenumber variation is induced as compared to the spherical inclusion shape. For oblate
quartz inclusions, the effect is more significant, when *c/a*=0.5 ca. 0.8 cm[-1] wavenumber variation for the 464 cm[-1] band is induced compared to the reference spherical inclusion case. We also show that it is possible to fit an effective ellipsoid to obtain a proxy for the averaged residual stress/strain within faceted inclusion. The difference between the volumetrically averaged stress of a faceted inclusion and the analytically calculated stress from the best-fitted effective ellipsoid is calculated to obtain the root mean square deviation (RMSD) for quartz, zircon, rutile, apatite and diamond inclusions in
garnet host. Based on the results of 500 randomly generated (a wide range of aspect ratio and random crystallographic orientation) faceted inclusion, we show that the volumetrically averaged stress serves as an excellent stress measure and the associated RMSD is less than 2%, except for diamond with a systematically higher RMSD (ca. 8%). This expands the applicability of the analytical solution for any arbitrary inclusion shape in practical Raman measurements.



## 1. Introduction

Raman elastic thermobarometry has been extensively used to recover the pressure and temperature (*P-T*) conditions of mineral inclusion entrapment, e.g. the mostly studied quartz-in-garnet inclusion-host pair (Ashley et al., 2014; Bayet et al., 2018; Enami et al., 2007; Gonzalez et al., 2019; Kouketsu et al., 2014; Taguchi et al., 2016, 2019; Zhong et al., 2019). Recently, quartz-in-garnet elastic barometry has been calibrated with experiments by synthesizing almandine garnets and quartz inclusions at high *P-T* conditions and comparing the entrapment pressure recovered based on residual pressure measured in quartz with the pressure applied in experiments (Bonazzi et al., 2019; Thomas and Spear, 2018). In practice, most mineral inclusions, e.g. quartz, zircon and rutile, are elastically anisotropic, and the associated effect needs to be addressed for better constraining the entrapment *P-T* conditions. Existing mechanical models for elastic thermobarometry typically assume the case of a spherical isotropic inclusion entrapped in an infinite isotropic host (e.g. Angel et al., 2017b; Gillet et al., 1984; Guiraud and Powell, 2006; Rosenfeld and Chase, 1961; Zhang, 1998). In recent studies, finite element (FE) simulations were applied to study anisotropic inclusions entrapped in cubic hosts such as garnet (Alvaro et al., 2020; Mazzucchelli et al., 2019). In this approach, the residual strain preserved within a mineral inclusion is related to the stress/strain state of the system upon entrapment via a relaxation tensor (*R*) that needs to be pre-calculated using the FE method or other numerical techniques (Mazzucchelli et al., 2019).

For an ellipsoidal, elastically anisotropic inclusion entrapped in an infinite isotropic host, an exact closed-form analytical solution is available for a long time (Eshelby, 1957; Mura, 1987). This solution has been widely applied to the earth science for many problems, such as viscous creep around inclusions (Freeman, 1987; Jiang, 2016); flanking structures (Exner and Dabrowski, 2010); elastic stress of inclusions at various scales (Meng and Pollard, 2014); microcracking and faulting (Healy et al., 2006), magma chamber induced deformations (Bonaccorso and Davis, 1999) etc. The advantage of such form is that no numerical software or programming is required and the solution can be obtained rapidly and precisely, with no numerical approximation error. The rapid calculation also permits in-depth, systematic stress and strain analysis of inclusion-host system or potentially Monte-Carlo simulation for uncertainty propagation. The procedure of calculating the residual stress in an ellipsoidal anisotropic inclusion embedded in an elastic, isotropic host is based on the equivalent eigenstrain method and the classical Eshelby solution (Eshelby, 1957; Mura, 1987). Recently, the Eshelby's solution has been applied to exhumed mineral inclusion entrapped in a host and the result is compared to the finite element method (Morganti et al., 2020). Mineral inclusions were measured for their crystallographic orientation and shape via X-ray diffraction and tomography (Morganti et al., 2020), but the significance of the aspect ratio, shape and crystallographic orientation have not been studied in a systematic way. More importantly, the Eshelby's solution only applies to perfectly ellipsoidal inclusions but natural mineral inclusions are faceted. Therefore, the uncertainty and limitation of using the Eshelby's solution to natural faceted inclusions remain to be investigated. In this study, we attempt to explore in-depth the Eshelby's solution to inclusion-host problem. A general analytical form is first presented in this paper (previous submission record available in Acknowledgements) following the Eshelby's equivalent eigenstrain method (Mura, 1987, chapter 4) to calculate residual stress and strain of an





anisotropic inclusion in an isotropic host. For inclusions belonging to certain crystallographic symmetry, such as tetragonal, hexagonal and trigonal, simplified explicit expressions describing residual stress and strain are derived. The analytical

formulas are cross validated against the numerical results obtained using a self-developed finite element code. Convergence tests are successfully performed to show the correspondence of the numerical (FE) and analytical solution. In-depth analysis of the effects due to: 1) inclusion elastic anisotropy, 2) inclusion aspect ratio, 3) relative orientation between the inclusion crystallographic and geometrical principal axis, are performed to show how they affect the application of elastic thermobarometry. The MATLAB code has also been made available together with the submission.

One major problem of using the Eshelby's solution to mineral inclusion is that natural inclusions are faceted in shape, which leads to a heterogeneous residual stress field (e.g. Chiu, 1978; Mazzucchelli et al., 2018). To resolve this issue, we use our self-developed 3D finite element code to simulate the residual stress distribution within faceted inclusions of varying shapes. Fitting an arbitrary 3D shape with effective ellipsoid is a common practice in image analysis and microstructural research (e.g. Ghosh and Dimiduk, 2011). We explore the possibility of using an effective ellipsoid to approximate the shape of a

faceted inclusion. The residual stress obtained from the analytical solution based on the best-fitted effective ellipsoid is used as a proxy to represent the volumetrically averaged stress within the faceted inclusion. By inspecting the numerical (FE) and analytical solutions, we have found that for most mineral inclusions, e.g. quartz, zircon, apatite and rutile, the volumetrically averaged stress represents very well the stress state of arbitrarily faceted inclusions. This may potentially provide useful guides to the future applications of elastic thermobarometry for any natural faceted mineral inclusions.

## 2. Method

### 2.1 Anisotropic inclusion embedded in isotropic host

We consider an anisotropic, ellipsoidal solid inclusion entrapped in an isotropic, infinite host at high *P-T* conditions. For a fully entrapped spherical inclusion, the assumption of an infinite host is justified when the distance between the inclusion and host grain outer boundaries, such as the thin-section surface, is more than 3 times the inclusion radius (Mazzucchelli et

al., 2018; Zhong et al., 2018). The principal axes of the inclusion are aligned along the Cartesian coordinates *x*, *y* and *z*, and their lengths are arbitrary. Upon entrapment at depth, it is considered that the inclusion and host are subject to the same stress field. The entrapment stress may be either hydrostatic or non-hydrostatic but it is treated to be homogeneous during the inclusion growth within the host grain or the overgrowth of the inclusion by the host grain. At this stage, the lattice strains of the inclusion and host are denoted by $\varepsilon_i^{incl}$ and $\varepsilon_i^{host}$, which are measured with respect to the reference room conditions. The

Voigt notation is applied here (Voigt, 1910). The entrapment lattice strains $\varepsilon_i^{incl}$ and $\varepsilon_i^{host}$ incorporate both pressure (compressibility) and temperature (thermal expansivity) effects. They can be obtained by relating the lattice parameters measured at high *P-T* conditions to their reference values under room *P-T* condition. For inclusions of cubic, tetragonal and orthorhombic symmetry classes, the three crystallographic axes *a*, *b* and *c* are mutually perpendicular to each other, so that the lattice strain $\varepsilon_i^{incl}$ can be readily expressed as:



$$\varepsilon_{xx}^{incl} = \frac{a}{a_0} - 1$$

$$\varepsilon_{yy}^{incl} = \frac{b}{b_0} - 1 \tag{1}$$

$$\varepsilon_{zz}^{incl} = \frac{c}{c_0} - 1$$

where e.g. $a_0$ is the reference lattice parameter measured at room conditions and $a$ is the lattice parameter measured at entrapment conditions. Note that for hexagonal and trigonal minerals (e.g. quartz), if the symmetry of lattice parameters is maintained at entrapment conditions (e.g. for quartz we keep $a=b$, $\alpha = \beta = 90^o$, $\gamma = 120^o$), Eq. 1 still holds. For lower symmetry systems with non-orthogonal crystallographic axes (triclinic and monoclinic systems), it's not possible to align all the crystallographic axes parallel to the Cartesian coordinates and the angles of $\alpha$, $\beta$ and $\gamma$ may also change at entrapment

condition compared to reference room condition. Therefore, transformation is needed to convert strains from the crystallographic axes in a unit cell to the Cartesian coordinate system for modelling the mechanical interaction between the inclusion and host. This can be done by using existing software such as PASCal (Cliffe and Goodwin, 2012), Win_Strain (http://www.rossangel.com/home.htm), and STRAIN (Ohashi and Burnham, 1973). A self-written MATLAB code is provided also in the Appendix following the Ohashi's method (Ohashi and Burnham, 1973) to calculate the lattice strain

based on the changes of lattice parameters. The results are the same with all existing software. For the case of an isotropic host under hydrostatic entrapment stress, the initial (entrapment) strain is expected to be isotropic and the principal strain components are simply one third of the volumetric strain, which can be directly obtained from the *PVT* relationship.

To simulate the exhumation of the inclusion-host system to room *P-T* conditions, we first unload the system by applying the strain opposite to the initial host strain state, i.e. $-\varepsilon_i^{host}$ (Fig. 1B), a procedure which leads to a stress- and strain-free host at

room conditions. This is an intermediate step that ignores elastic interaction between the inclusion and host, and the inclusion will possess a virtual strain $\varepsilon_i^{incl} - \varepsilon_i^{host}$, as the internal inclusion-host boundary experiences the unloading strain $-\varepsilon_i^{host}$. At this moment, the stress state of the inclusion can be readily expressed using the linear-elastic constitutive law as: $C_{ij}^{incl}(\varepsilon_j^{incl} - \varepsilon_j^{host})$, where $C_{ij}^{incl}$ is the elastic stiffness tensor of the inclusion at room *T*. Einstein summation is used. It is straightforward to note that mechanical equilibrium is not satisfied at this intermediate step because, in general, there is a

stress jump between the stressed inclusion and the stress-free host. Using the proposed approach, solving the original mechanical problem is reduced to superposing the homogeneous unloading strain field $-\varepsilon_i^{host}$ with a non-uniform solution for an initially stressed and strained inclusion embedded in a stress and strain free host at room *T*. The latter one is practically an eigenstrain problem (Eshelby, 1957). The thermal effects on the elastic stiffness tensor $C_{ij}^{incl}$ have no influence on the superposed deformation field driven by the eigenstrain load due to the mismatch between the lattice strains of the

inclusion and the host. The stress and strain of the inclusion that serve as driving force for elastic interaction are as follows:





$$\varepsilon_i^* = \varepsilon_i^{incl} - \varepsilon_i^{host} \qquad\qquad (2)$$

$$\sigma_i^* = C_{ij}^{incl}(\varepsilon_j^{incl} - \varepsilon_j^{host})$$

where $\varepsilon_i^*$ are referred to as inclusion eigenstrains and $\sigma_i^*$ are eigenstresses (Mura, 1987). The eigenstresses correspond to the stress that a soft inclusion would experience if it was perfectly confined by a stiff host, i.e. the host was not allowed to deform elastically. The eigenstrain and eigenstress are the functions of the lattice strain of inclusion and host at entrapment conditions (taking room conditions as reference state) and the stiffness tensor of the inclusion at room *P-T* condition.

Because mechanical equilibrium is not satisfied for the stressed inclusion embedded in a stress-free host, elastic deformation will occur (stage shown in Fig. 1b to Fig. 1c). The amount of elastic deformation that affects the inclusion with a pre-strain $\varepsilon_i^{incl} - \varepsilon_i^{host}$ in a stress-free host to mechanical equilibrium is denoted as $\varepsilon_i$. The strain $\varepsilon_i$ is shown in Fig. 1b to 1c, and the pre-strained state with strain $\varepsilon_i^{incl} - \varepsilon_i^{host}$ is taken as the reference state for this elastic deformation field. By adding the strain $\varepsilon_i$ to the inclusion, we obtain the final residual stress and strain state as follows:

$$\varepsilon_i^{res} = \varepsilon_i^{incl} - \varepsilon_i^{host} + \varepsilon_i \qquad\qquad (3)$$

$$\sigma_i^{res} = C_{ij}^{incl}(\varepsilon_j^{incl} - \varepsilon_j^{host} + \varepsilon_i)$$

where $\varepsilon_i^{res}$ and $\sigma_i^{res}$ are the final residual strains and stresses of the inclusion, which are the true physical stress and strain the inclusion experiences. This final stage is shown in Fig. 1c. Finding the strain $\varepsilon_i$ will solve the anisotropic inclusion problem. This will be sought in the next section using the equivalent eigenstrain method and Eshelby's solution.

**2.2 Solving the problem with Eshelby's solution**

The Eshelby's solution treats a homogeneous, ellipsoidal, isotropic inclusion embedded in an infinite isotropic host (Eshelby,
1957). Following Eshelby (1957), we replace the inclusion-host system by an isotropic homogeneous space with elastic tensor $C_{ij}^{host}$ and load the ellipsoidal inclusion region with an equivalent eigenstrain $e_i^*$ (to differentiate from the previously introduced eigenstrain term $\varepsilon_i^*$ due to lattice strain mismatch). Without elastic interaction, the inclusion would experience a stress $-C_{ij}^{host}e_j^*$ under perfect confinement (e.g. a positive eigenstrain (expansion) leads to compressive stress, which is negative). After elastic interaction, the inclusion strain and stress under mechanical equilibrium due to a constant eigenstrain
(eigenstress) load applied to an ellipsoidal region of otherwise homogeneous elastic space, can be expressed as follows (Eshelby, 1957; Mura, 1987):

$$\varepsilon_i^{res} = S_{ij}e_j^* \qquad\qquad (4)$$

$$\sigma_i^{res} = C_{ij}^{host}(S_{jk}e_k^* - e_j^*)$$





where $S_{ij}$ is the Eshelby's tensor, which gives the one-to-one mapping between the equivalent eigenstrain ($e_j^*$) and a homogeneous residual strain ($\varepsilon_i^{res}$) within the inclusion region. The Eshelby's solution is manifested in this tensor, which is only a function of the inclusion shape and the Poisson ratio ($\nu$) of the isotropic host. For a spherical inclusion, $S_{ij}$ is

symmetric and can be significantly simplified as follows:

$$S_{11} = S_{22} = S_{33} = \frac{7-5\nu}{15(1-\nu)}$$

$$S_{12} = S_{23} = S_{13} = \frac{5\nu-1}{15(1-\nu)}$$ 

$$\tag{5}$$

$$S_{44} = S_{55} = S_{66} = \frac{4-5\nu}{15(1-\nu)}$$

All the other components are zero. For general ellipsoidal inclusions, the $S_{ij}$ tensor is given in the Appendix and a MATLAB script for calculating the $S_{ij}$ tensor is provided in the supplementary materials (see Appendix for more details for using the script).

Following the equivalent eigenstrain method (Mura, 1987, chapter 4), one may appropriately choose the equivalent

eigenstrain $e_j^*$ to let $S_{ij}e_j^*$ in Eq. 4 equals the strain $\varepsilon_i$ in Eq. 3 that drives the pre-strained anisotropic inclusion into mechanical equilibrium with stress-free isotropic host, i.e. we have:

$$\varepsilon_i = S_{ij}e_j^* \tag{6}$$

By doing so, the stresses in the original anisotropic heterogeneity and the equivalent isotropic inclusion will be equal. This is because the host is stressed (and strained) by the same amount following Eq. 6, which leads to the same inclusion stress because the traction is matched between inclusion and host. By replacing the strain $\varepsilon_i = S_{ij}e_j^*$ into $\sigma_i^{res}$ in Eq. 3 and equating

the stresses $\sigma_i^{res}$ in Eq. 3 with Eq. 4, we obtain the following relation:

$$\sigma_i^{res} = C_{ij}^{incl}\big(\varepsilon_j^{incl} - \varepsilon_j^{host} + S_{jk}e_k^*\big) = C_{ij}^{host}(S_{jk}e_k^* - e_j^*) \tag{7}$$

The equivalent eigenstrain $e_k^*$ can be solved from this equation. By substituting the obtained $e_k^*$ back into Eq. 7, we may concisely express the final result for the residual strain and stress of the anisotropic inclusion embedded in isotropic infinite host:

$$\varepsilon_i^{res} = (I_{ij} - M_{ij})\varepsilon_j^* \tag{8}$$

$$\sigma_i^{res} = C_{ik}^{incl}(I_{kj} - M_{kj})\varepsilon_j^*$$

where $I_{ij}$ is the identity matrix. The dimensionless matrix $M_{kj}$ can be expressed as follows:





$$M_{ij} = \left[ C_{ik}^{incl} - C_{il}^{host}(I_{lk} - S_{lk}^{-1}) \right]^{-1} C_{kj}^{incl} \tag{9}$$

The dimensionless matrix $M_{ij}$ depends on the elastic stiffness properties of the inclusion and the host as well as the aspect ratio of the inclusion manifested by the Eshelby's tensor. The components of this matrix are close to zero for a stiff host or a soft inclusion (no elastic relaxation so that $\sigma_i^{res} \rightarrow C_{ij}^{incl}\varepsilon_j^*$) and it approaches the identity matrix for an infinitely soft host (in this case, $\sigma_i^{res} \rightarrow 0$). An extreme case is represented by gas/liquid inclusion whose $C_{ij}^{incl}$ is low compared to the host stiffness, thus this dimensionless matrix $M_{ij}$ approaches zero and the isochoric assumption for the gas/liquid inclusion is

justified. The $M_{ij}$ matrix can be readily calculated by using the published elastic stiffness tensor at room $P$-$T$ conditions (e.g. Bass, 1995). A MATLAB script is given in the supplementary data to perform this task (details of using the code can be found in Appendix)

**2.3 Back-calculating eigenstrain terms based on residual inclusion strain**

The wavenumber shifts of Raman peaks are induced by lattice strain. By measuring wavenumber shift of the inclusion in a

thin-section, it is possible to recover the residual strain preserved within the inclusion (Angel et al., 2019; Murri et al., 2018). This can be done by using the Grüneisen tensor. Therefore, $\varepsilon_i^{res}$ can be obtained with e.g. least-square fitting method (Murri et al., 2018) and the residual stress can be readily expressed as $\sigma_i^{res} = C_{ij}^{incl}\varepsilon_j^{res}$.

By inverting the right-hand matrix in Eq. 8, the eigenstrain terms can be expressed as a function of residual strain $\varepsilon_i^{res}$:

$$\varepsilon_i^* = \left\{ I - (I_{jl} - S_{jl}^{-1})^{-1} C_{ij}^{host^{-1}} C_{lk}^{incl} \right\} \varepsilon_k^{res} \tag{10}$$

For tetragonal or hexagonal minerals, e.g. zircon, rutile and apatite, the stiffness tensor comprises six independent

components: $C_{11}^{incl}$, $C_{12}^{incl}$, $C_{13}^{incl}$, $C_{33}^{incl}$, $C_{44}^{incl}$, $C_{66}^{incl}$. For trigonal symmetry such as in the case of $\alpha$-quartz, another independent component $C_{14}^{incl} = -C_{24}^{incl}$ is also present. For all these mineral inclusions, and for axially symmetric residual strains, $\varepsilon_i^*$ can be significantly simplified as follows ($\varepsilon_x^* = \varepsilon_y^* \neq \varepsilon_z^*$):

$$\varepsilon_x^* = \varepsilon_x^{res} + \frac{5(1-\nu)}{2(7-5\nu)}\left[ (\bar{C}_{11}^{incl} + \bar{C}_{12}^{incl})\varepsilon_x^{res} + \bar{C}_{13}^{incl}\varepsilon_z^{res} \right] - \frac{3-5\nu}{4(7-5\nu)}(2\bar{C}_{13}^{incl}\varepsilon_x^{res} + \bar{C}_{33}^{incl}\varepsilon_z^{res}) \tag{11}$$

$$\varepsilon_z^* = \varepsilon_z^{res} - \frac{3-5\nu}{2(7-5\nu)}\left[ (\bar{C}_{11}^{incl} + \bar{C}_{12}^{incl})\varepsilon_x^{res} + \bar{C}_{13}^{incl}\varepsilon_z^{res} \right] + \frac{13-15\nu}{4(7-5\nu)}(2\bar{C}_{13}^{incl}\varepsilon_x^{res} + \bar{C}_{33}^{incl}\varepsilon_z^{res})$$

where $\bar{C}_{ij}^{incl} = C_{ij}^{incl}/G$ is the dimensionless inclusion stiffness tensor scaled by the host shear modulus. Interestingly, for trigonal $\alpha$-quartz inclusions ($C_{14}^{incl} \neq 0$), the stiffness tensor component $C_{14}^{incl}$ is not present in the expression given by Eq. 11.

In fact, the terms in the brackets are simply the residual stress components:





$$\varepsilon_x^* = \varepsilon_x^{res} + \frac{5(1-\nu)}{2G(7-5\nu)}\sigma_x^{res} - \frac{3-5\nu}{4G(7-5\nu)}\sigma_z^{res} \qquad (12)$$

$$\varepsilon_z^* = \varepsilon_z^{res} - \frac{3-5\nu}{2G(7-5\nu)}\sigma_x^{res} + \frac{13-15\nu}{4G(7-5\nu)}\sigma_z^{res}$$

By substituting the stiffness tensor components and the measured residual strains, the eigenstrains can be directly calculated. The equation above thus allows estimation of the entrapment (hydrostatic or non-hydrostatic) stress (or strain) conditions by known the residual stress and strain conditions of the inclusion.

### 3. Cross validation against finite element solution

We have validated our implementation of the proposed analytical framework against independent finite element (FE) solutions. A self-written 3D FE code is used to validate the presented analytical solution (Dabrowski et al., 2008; Zhong et al., 2018). For validation purposes, we used spheroidal quartz inclusions in an almandine garnet host. Adaptive tetrahedral computational meshes, with the highest resolution within and around the inclusion, are generated with *Tetgen* software (Si, 2015). The anisotropic elastic properties of quartz inclusion at room *T* are based on Heyliger et al. (2003). The host garnet

elasticity is first treated as isotropic based on Jiang et al. (2004). The model length is set as 10 times (denoted as *10 below) the inclusion's diameter (for spheroidal inclusion case, the model domain is a box where the side lengths are proportional to the corresponding axes lengths of the inclusion). For the model validation, we adjust the eigenstrain term to generate precisely 1 GPa compressive residual stress for spherical inclusion in infinite garnet host. This is done by letting $\sigma_{xx}^{res} = \sigma_{yy}^{res} = \sigma_{zz}^{res} = -1$ GPa hydrostatically and substituting $\sigma_i^{res}$ into Eq. 8 to back-calculate the eigenstrain $\varepsilon_j^*$. Then, the

spheroidal inclusion is loaded by the calculated eigenstrain $\varepsilon_j^*$ in the FE code, and the residual stress is compared to results obtained by the presented analytical solution. The choice of eigenstrains (either loading the inclusion to 1 GPa pressure or any other residual stress value) is not influential for the validation purposes, as long as both the analytical and FE methods take the same eigenstrains. This and other successful, more general tests with arbitrary aspect ratio and eigenstrains, have been performed but are not reported here.

In Fig. 2a, the numerically and analytically obtained residual stresses are plotted together as a function of the aspect ratio of the tested spheroidal inclusions. In Fig. 2b, the difference is plotted as a function of element count and boundary distance (*5, *10 and *20). It is clearly shown that the two sets of solutions converge with increasing the number of mesh elements and the computational box size. The success of this convergence test validates the correctness of our presented analytical model (also FE code) for an anisotropic ellipsoidal inclusion entrapped in an isotropic space.

In addition, we have also tested the effect of applying cubic elastic stiffness tensor of almandine from Jiang et al. (2004) and compared the residual stress with the results obtained for an elastically isotropic garnet (blue dots in Fig. 2a). The difference in residual stresses obtained with FE method using anisotropic garnet host and the analytical solution (implicitly assuming isotropic host) is less than 0.001 GPa. This suggests that it is not necessary in practice to consider the anisotropy of garnet





host. This has been also reported in e.g. Mazzucchelli et al. (2019). It was suggested that the elastic anisotropy of cubic
garnet has no significant impact on the result of elastic barometry. Thus, effective isotropic elastic properties of garnets may
be used to model the inclusion-host elastic interaction.

## 4. Model applications

### 4.1 Effect of inclusion aspect ratio on residual stresses

In Eq. 8, the aspect ratio of the inclusion only affects the Eshelby tensor. Here, we choose some common inclusions in
metamorphic garnets, including quartz, apatite, zircon and rutile as examples to test the effect of inclusion aspect ratio on
residual stresses. The data sources of elastic stiffness tensors of the studied minerals are listed in the caption of Fig. 3. Here,
we first focus on spheroidal inclusions, and let the crystallographic *c*-axis coincide with the geometrical *z*-axis of the
inclusion. The inclusion aspect ratio is controlled by varying the lengths of the principal *z*- and *x*-(*y*)-axes, and it is denoted
by *c/a* ratio for simplicity in the following text (note it is not the ratio of the lattice parameters *c* and *a*).

To isolate the effect of the inclusion aspect ratio, the eigenstrains for various inclusion minerals are all set to create 1 GPa
compressive hydrostatic residual stress for the reference spherical inclusion embedded in an isotropic, infinite garnet host,
which is the same approach as in the previous "cross-validation" section. Therefore, the stress variations shown in Fig. 3
only represent the mechanical effect purely due to varying the geometrical aspect ratio *c/a*, and they allow direct comparison
among different common mineral inclusions in metamorphic garnets.

Among all the tested minerals, the residual stress in quartz inclusions is the least sensitive to variations in aspect ratio. For
prolate quartz inclusion (*c/a*>1), the change of $\sigma_x^{res}$ due to shape variation is within 0.1 GPa and the change of $\sigma_z^{res}$ is within
0.2 GPa. The effects of varying the aspect ratio of prolate zircon and apatite inclusions are similar, with variation of $\sigma_x^{res}$
from the reference spherical case reaching up to ca. 0.12 GPa and $\sigma_z^{res}$ up to ca. 0.2 GPa.

The residual stress in rutile is the most sensitive to aspect ratio ariations. With increasing *c/a* ratio from 1, $\sigma_x^{res}$ in prolate
rutile inclusions increases up to ca. 0.2 GPa and $\sigma_z^{res}$ decreases by ca. 0.6 GPa. This is relevant for practical measurement as
rutile often forms needle-shaped crystals.

The pressure (negative mean stress) is significantly less sensitive to inclusion aspect ratio variations. For prolate inclusions
of quartz, zircon and apatite, the residual pressure differs from the reference level (spherical inclusion shape) by only ca.
0.01 GPa when the aspect ratio *c/a* is stretched up to 4. For oblate inclusion (*c/a*<1), the pressure variation is typically below
0.1 GPa.

The residual stress in mineral inclusions can be easily converted into residual strain, which can be directly translated into
Raman shifts (Angel et al., 2019; Murri et al., 2018). The effects of varying the aspect ratio of a quartz inclusion on Raman
wavenumber shifts (see Fig. 4) are determined using the calculated residual strain components and the Grüneisen tensor
(Murri et al., 2018). The same model settings are applied, with 1 GPa compressive hydrostatic residual stress characterizing





the case of a spherical quartz inclusion. It is shown that for prolate inclusions, aspect ratio only introduces minor effects on the Raman shifts. For example, varying the *c/a* ratio between 1 and 5 induces a wavenumber variation of less than 0.3 cm$^{-1}$ for the 464 cm$^{-1}$ Raman peak. This is in most cases insignificant from the viewpoint of practical Raman measurements. The *b/a* ratio variations are also shown to be insignificant for the spectral shifts, with changes less than 0.2 cm$^{-1}$ for the 464 cm$^{-1}$ peak. For oblate inclusion, the impact of inclusion shape is shown to be more significant. For the 464 cm$^{-1}$ peak, the change

of wavenumber shift can reach 0.8 cm$^{-1}$ for strongly oblate inclusion (*c/a*=0.25), and it is ca. 0.3-0.4 cm$^{-1}$ for less oblate inclusion (*c/a*=0.5).

Our results show good consistency with the Raman data reported in Kouketsu et al. (2014), where quartz inclusions with different aspect ratio were measured and no significant variation on spectral shift was found.

### 4.2 Effect of inclusion crystallographic orientation with respect to long-axis

In nature, the crystallographic axes of an inclusion are not necessarily aligned parallel to its geometrical axes. In this section, the effect of varying the crystallographic orientation with respect to the geometrical axes on the resulting Raman wavenumber shift is systematically studied using the proposed analytical model. Here, we reorient the crystallographic *c*-axis of a spheroidal quartz inclusion from its long axis (Fig. 5) between 0 (crystallographic *c*-axis parallel to the geometrical long axis) and 90 degrees (*c*-axis perpendicular to the long axis). The same eigenstrain is applied for quartz as from previous

section. The elastic stiffness tensor of quartz is from Heyliger et al., (2003) and of isotropic almandine garnet is from Milani et al., (2015). The predicted Raman spectral shifts are calculated based on the Grüneisen tensor calibrated by Murri et al. (2018).

The results are shown in Fig. 5. For an aspect ratio of 2, the 464 cm$^{-1}$ band varies ca. $\pm$0.2 cm$^{-1}$ when the orientation of the crystallographic c-axis is varied between the long and the short geometrical axis. The effect of crystallographic orientation

(*c*-axis) on the Raman shifts increases towards higher aspect ratio. For an aspect ratio of 5, the 464 cm$^{-1}$ band varies ca. $\pm$0.4 cm$^{-1}$. Similarly, for 206 and 128 cm$^{-1}$ bands, the maximal wavenumber variations compared to the reference case of a spherical inclusion are ca. 0.8 and 0.3 cm$^{-1}$, respectively. The results suggest that the crystallographic orientation of a quartz inclusion with respect to its geometrical axes has no significant impact on the predicted Raman spectral shift, as long as the geometrical aspect ratio is not higher than 2-3.

### 4.3 Effect of faceted inclusion shape

The analytical solution presented in this study is derived for an ellipsoidal anisotropic inclusion. However, the shape of a natural mineral inclusion may exhibit corners, edges and facets, which results in stress concentration effects and may have an impact on the overall level of the residual stress.

Here, we explore the possibility of using an effective ellipsoid to fit the shape of a faceted inclusion. We use the equivalent

aspect ratio to calculate the residual stress/strain based on the presented analytical solution for ellipsoidal inclusions. Fitting



an arbitrary, irregular shape using an ellipsoid in 3D (or an ellipse in 2D) is a common practice in image analysis (Chaudhuri and Samanta, 1991; Li et al., 1999). A pixelated 3D image is used to calculate the second-order moment of the object shape to minimize the mismatch between the 3D irregular inclusion shape and the effective ellipsoid. The method allows for obtaining the lengths and orientations of the major, minor and intermediate axes of the effective ellipsoid (method described

in Appendix and a MATLAB code is provided in supplementary data to perform this task).

Similar to previous sections, we use the eigenstrain components that can load the reference spherical inclusion of any given mineral embedded in isotropic almandine garnet host into 1 GPa compressive residual stress. The tested inclusion shapes include: cylinder, tetrahedron, cuboid, octahedron, hexagonal prism, and icosahedron. To vary the aspect ratio, the inclusion shape is stretched in the $z$-axis direction, which is parallel to the crystallographic $c$-axis as shown in Fig. 6.

We study five inclusion minerals: quartz (elastic tensor from Heyliger et al., 2003), zircon (Bass, 1995), rutile (Wachtman et al., 1962), fluorapatite (Sha et al., 1994), and diamond (Bass, 1995). Almandine garnet is taken as the host grain (Milani et al., 2015). For each FE mesh, the size of the computational box is set more than 10 times the inclusion size and adaptive mesh is generated with highest mesh resolution within and close to the inclusion. 10-node tetrahedron elements with quadratic (second-order) shape (interpolation) functions for the displacement field are used. In total, there are ca. 2 million

elements per model (numerical error less than 0.0003 based on benchmark results in Fig. 2).

The results of numerical simulations are shown in Fig. 6. The effective aspect ratio for all different inclusion shapes together with the residual stress components are given in supplementary data. The residual stress in non-ellipsoidal inclusions based on FE model is heterogeneous and we monitor the stress state: 1) at the centroid (shorten as CT) point (red dots in Fig. 6), 2) as the volumetric average (VA) within the entire inclusion (orange dots)

The root mean square deviation (RMSD) is calculated by comparing the residual stress from the finite element solutions based on various stress evaluation scheme (CT/VA) and analytical model using the best-fitted effective aspect ratio (Table 1). It is clearly shown that the VA stresses of quartz, zircon, rutile and apatite inclusion are remarkably similar to the analytical results, with RMSD generally lower than 0.02-0.03 GPa (ca. 2%). From CT to VA, a significant improvement on RMSD of a factor of 2 to 3 is obtained.

The only exception among the studied minerals is diamond, where the RMSD is higher than in the case of other inclusions, which are elastically softer. This is consistent with the high "geometrical correction factor" reported for diamond in Mazzucchelli et al. (2018). However, as an improvement from Mazzucchelli et al. (2018), where the geometrical correction factor must be applied for all inclusion phases to correct the residual stress due to shape effects, we have found that the stress variation due to varying inclusion shape for minerals such as quartz, zircon, apatite and rutile can be satisfactorily

approximated by using the proposed approach of the equivalent ellipsoidal inclusion, with RMSD generally lower than 3-4% for most of the studied inclusion shapes. To achieve this improved and satisfactory level of prediction: 1) we have used best-fit ellipsoids to better approximate inclusion shapes, instead of a crude measure of the length/width ratio of e.g. cuboidal or





cylindrical inclusion; 2) we have considered not only the centroid point of the inclusion (which indeed yields a larger RMSD), but also the volumetric average (VA) for the residual stress state sampled during stress measurements, which
interestingly provides a significantly better approximation for the residual stress/strain state of the tested mineral inclusions. This is practical and useful in Raman measurement because it is possible to perform either 1) multiple-point averaging during Raman analysis within the entire inclusion, or 2) defocus the laser beam to take into account a larger volume for the inclusion strain heterogeneity. For tiny inclusions (size of ca. 1~3 μm) and for a typical in-plane laser beam diameter of ca. 1~2 μm, the stress/strain averaging is, in fact, implicitly performed during measurements,. Based on FE analysis, it is clearly
shown that the volumetrically averaged stress within the inclusion, rather than centroid point measurements, may provide a closer match compared to the stress predicted based on the presented analytical solution developed for the best-fitted ellipsoid. This effect becomes statistically more significant when the faceted inclusion shape and crystallographic orientation are independent as demonstrated in the next section.

### 4.4 Irregularly faceted shapes and random crystallographic orientation

In nature, the shape of mineral inclusions is not necessarily highly symmetric as treated in the previous section and the crystallographic orientation can be generally random with respect to the principal geometric axes of the inclusion. Here, a MATLAB script is used to generate completely random 3D inclusion shapes by prescribing random vertices (non-coplanar 5 to 24 vertices) and connecting them to form a closed 3D shape. Delaunay triangulation is used to form 3D volumes enclosed by the triangular faces. "*Tetgen*" software is again used to generate unstructured tetrahedron computational meshes fitted to
the inclusion surface. The effective aspect ratio (geometrical longest to shortest axis of the best-fitted effective ellipsoid) is controlled to be within 6. In total, we have generated ca. 500 random 3D inclusion shapes and performed finite element simulation for the previously studied set of anisotropic inclusion minerals (quartz, zircon, rutile, apatite and diamond) to calculate the elastic stress field. The generated random shapes are plotted in supplementary data (see the *.gif* animation to illustrate 100 selected examples of 3D inclusion shapes). We further allow the crystallographic *c*-axis to be pointing along
the orientation randomly chosen parallel to either the longest, intermediate or shortest geometrical axis (best-fitted using the method of Chaudhuri and Samanta, 1991). The FE results are compared to the analytical results based on the effective ellipsoidal inclusion with the same crystallographic orientation. This Monte-Carlo type FE simulation allows us to investigate how much stress deviation can be generated for irregularly faceted inclusion shapes with random crystallographic orientation, and how accurate the analytical approach based on the best fitted ellipsoid is to describe the residual stress state
in an irregularly shaped inclusion, depending on the stress sampling scheme (CT/VA). The results are plotted in Fig. 7 (raw data of FE simulations can be found in supplementary data).

For centroid point (CT), quartz inclusions have the lowest RMSD of ca. 0.03 GPa for all the three normal stress components and diamond inclusions have the highest RMSD of ca. 0.11 GPa. In general, CT stress shows a systematically higher RMSD than VA stresses. When the stress is volumetrically averaged within the inclusion (VA), the RMSD dramatically drops to a
nearly perfect match between the FE results for irregularly faceted inclusion and the analytical prediction based on the best




fitted ellipsoid. The RMSD of volumetrically averaged residual stresses (VA) of quartz, zircon, rutile and apatite are all lower than ca. 0.02 GPa (2%) and it shows no obvious dependence on the effective aspect ratio even for the extremely elongated or flattened inclusions (see the near-perfect alignment of the orange dots and 1-to-1 ratio line in the middle and bottom panel of Fig. 7).

Thus the volumetric average of the residual stress within the inclusion provides a sufficiently reliable match between the exact results for irregularly shaped inclusions and the approximate predictions based on the analytical solution. This shows that it is possible to approximate the stress/strain state of the inclusion using an effective ellipsoid shape for the tested inclusions including quartz, zircon, rutile and apatite. Only diamond has a notably higher level of RMSD, exhibiting a systematic discrepancy between the exact numerical and approximate analytical results. This indicates that using the

proposed equivalent analytical model for diamond inclusion may lead to a potential overestimation of the residual stress by ca. 0.07 GPa (7%). However, this RMSD may still be acceptable as a crude estimate or may serve as an upper limit for elastic thermobarometry.

### 5. Non-linear elasticity at room $T$

The presented model builds on a linear elastic constitutive law at room temperature, i.e. $\sigma_i^{res} = C_{ij}^{incl} \varepsilon_j^{res}$. This assumption is

appropriate when the residual stresses/strains of the inclusion are low, thus the application of a constant anisotropic stiffness tensor $C_{ij}^{incl}$ determined at room $P$-$T$ conditions introduces no significant errors. For highly stressed mineral inclusions, e.g. inclusions in diamond host from mantle xenoliths where the residual inclusion pressure may reach several GPa, this approximation may lead to non-negligible deviations. To eliminate such error, the stiffness tensor $C_{ij}^{incl}$ needs to be treated as a function of either non-hydrostatic stresses or anisotropic strains, i.e. $C_{ij}^{incl}(\sigma_i^{res})$ or $C_{ij}^{incl}(\varepsilon_i^{res})$. In experimental studies,

$C_{ij}^{incl}$ is often described as a function of hydrostatic pressure, e.g. Bass (1995). It is beyond the scope of this paper to develop a method of fitting $C_{ij}^{incl}$ with respect to the individual stress tensor components. If the stiffness tensor $C_{ij}^{incl}$ can be parameterized by $\sigma_i^{res}$ or simply as a function of pressure as a first-order approximation, the residual stresses/strains are readily calculated by iterating Eq. 8, while updating the $C_{ij}^{incl}$ tensor using the calculated inclusion stress or strain during the iteration loop. Thus, the developed analytical method based on the Eshelby's solution can be extended to the case of a non-

linear inclusion phase as long as $C_{ij}^{incl}$ can be parametrized in terms of stress or strain components, or their invariants.

### 6. Concluding remarks and petrological implications

In this study, we use the classical Eshelby solution combined with the equivalent eigenstrain method to calculate the residual strain and stress in an anisotropic, ellipsoidal mineral inclusion embedded in an elastically isotropic host. The residual stresses can be expressed by a linear operator (Eq. 8) acting on the eigenstrain. The linear operator depends on the

anisotropic elastic stiffness tensor of the inclusion evaluated at room P-T conditions, the shape of the inclusion, and the shear





modulus and Poisson ratio of the host. The studied mechanical problem is loaded by an eigenstrain term, which is given by the difference between the lattice strains of the inclusion and host at the *P–T* conditions of entrapment.

The effect of inclusion aspect ratio on the inclusion residual stress and strain has been investigated quantitatively. The residual stress in quartz inclusions exhibits the least sensitivity to aspect ratio changes and rutile shows the most pronounced

variation. The popularly used quartz-in-garnet system is studied in more details. For prolate quartz inclusions, the residual stress variations caused by varying inclusion shape are shown to be insignificant when the crystallographic *c*-axis is subparallel to the geometrical long axis. The Raman wavenumber variation is less than 0.4 cm$^{-1}$ for the 464 cm$^{-1}$ peak even for highly elongated inclusions with an aspect ratio of 5. For oblate quartz inclusions with an aspect ratio of ca. 0.5, the additional wavenumber shift may reach ca. 0.8 cm$^{-1}$. Therefore, it is useful in practice, although potentially technically

difficult, to have an estimate of the crystallographic *c*-axis orientation when studying highly stretched or flattened quartz inclusions. As long as the *c*-axis is sub-parallel to the geometrical long-axis, the additional wavenumber shifts due to the inclusion aspect ratio is minor.

Our proposed analytical procedures to model residual inclusion stress and strain state do not require pre-FE simulation to obtain the 6-by-6 "relaxation tensor" as proposed by Mazzucchelli et al. (2019). For application purposes, as long as the

lattice strains of inclusion ($\varepsilon_i^{incl}$) and host ($\varepsilon_i^{host}$) at high *P-T* conditions are available, it is possible to calculate the eigenstrain term by subtracting them following Eq. 2. Given the driving eigenstrain, the residual strain and stress preserved within an anisotropic, ellipsoidal inclusion in isotropic host can be easily calculated using Eq. 8. The proposed procedure can be inversely applied to retrieve the residual strain/stress of any natural mineral inclusions embedded in elastically isotropic hosts, such as garnets.

The presented model is only exact for perfectly ellipsoidal inclusions. In nature, inclusions often possess different shapes with facets and edges. Finite element simulations on various faceted inclusion shapes showed that the residual stress is modified to a different degree as compared to the simple ellipsoidal inclusion case, depending on the relative elastic properties between the inclusion and the host grain. However, the proposed approach of using the analytical result for the best-fitted effective ellipsoids yields remarkably good approximation for all the tested inclusion shapes. including highly

irregular. The RMSD comparing the FE numerical solution for faceted inclusion and the analytical solution based on effective best-fitted ellipsoid is typically less than 2% for quartz, zircon, apatite and rutile inclusions. The only exception are the elastically stiff diamond inclusions, where the RMSD reaches 7%. This finding expands the applicability of the analytical framework to arbitrarily shaped inclusions, whose elastic stiffness is not signifficantly higher than host (such as quartz, rutile, zircon and apatite). One important petrological implication is that it is possible to take the volumetrically averaged

stress/strain within the inclusion and use it as a proxy to represent the residual stress/strain state of the inclusion. Then the proposed analytical framework may be used to recover the entrapment condition by back-calculating the eigenstrain using the volumetrically averaged residual stress/strain and the effective ellipsoid aspect ratio of the inclusion (Eq. 8). In fact,





averaging the stress/strain within a certain volume is implicitly done in practical Raman measurement, for example for tiny µm size inclusion with laser beam size typically exceeding 1 µm.

**Author contribution**

XZ and MD conceived the idea and developed the analytical method. XZ did the Finite Element model. XZ, MD and BJ wrote the manuscript together.

**Code availability**

All MATLAB codes have been uploaded as supplementary files. Details to use the code can be found in Appendix or directly contact via xinzhong0708@gmail.com.

**Data availability**

All calculated data are available as excel file with details provided in Appendix.

**Acknowledgements and Declarations**

This work has been finished during 2017~2018 and the paper has been submitted to American Mineralogist in Nov. 2018 (No. 6895) with two major rounds of reviews but was finally rejected after one year. Later it was submitted to Contributions to Mineralogy and Petrology in late 2019 and Lithos in 2020 with complete revision but was rejected potentially due to conflict of interest. We acknowledge the team of reviewers who have commented this manuscript. This project has been supported by the early postdoc mobility fellowship of Swiss National Science Foundation (P2EZP2_172220) and the Alexander von Humboldt fellowship to XZ, and the European Union's Horizon 2020 Research and Innovation Programme under the ERC Advanced Grant Agreement n°669972, 'Disequilibrium Metamorphism' ('DIME') to BJ. MD acknowledges the PGI-NRI statutory funds (Project No. 62.9012.2063)

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



**Figures and Tables**

**Table 1**. Root mean square deviation (RMSD) of finite element solution of symmetrically shaped non-ellipsoidal inclusion in Fig. 6 compared to the analytical solution of equivalent spheroidal inclusions. Isotropic almandine garnet is used as host. For each inclusion mineral and inclusion shape, the aspect ratio varies from 0.2~5. Effective aspect ratio is calculated for

each shape and used for the analytical solution to obtain the residual stress state. The inclusion is loaded by eigenstrain that creates 1 GPa hydrostatic residual pressure for spherical inclusion in infinite host. Thus, any stress variation can only be caused by shape change. The calculated stress data for each individual FE run is given in supplementary data. Stress is obtained for 1) the centroid point (CT), and 2) volumetric average (VA) of the entire inclusion (see Fig. 6 for illustration). The RMSD is calculated by comparing the FE results and analytical results based on the best-fitted effective ellipsoid. The

unit of RMSD is in GPa. Elasticity of inclusion mineral given in the caption of Fig. 6.

| Shape | Cylinder | | Tetrahedron | | Cuboid | | Hexagonal | | Octahedron | | Icosahedron | |
|---|---|---|---|---|---|---|---|---|---|---|---|---|
| Location | CT | VA | CT | VA | CT | VA | CT | VA | CT | VA | CT | VA |
| Quartz | 0.041 | 0.021 | 0.034 | 0.044 | 0.042 | 0.026 | 0.038 | 0.021 | 0.055 | 0.022 | 0.011 | 0.005 |
| Zircon | 0.045 | 0.023 | 0.042 | 0.048 | 0.112 | 0.028 | 0.047 | 0.024 | 0.084 | 0.017 | 0.028 | 0.006 |
| Rutile | 0.063 | 0.039 | 0.049 | 0.029 | 0.158 | 0.039 | 0.065 | 0.039 | 0.127 | 0.026 | 0.034 | 0.003 |
| Apatite | 0.047 | 0.029 | 0.052 | 0.057 | 0.049 | 0.035 | 0.045 | 0.029 | 0.062 | 0.024 | 0.014 | 0.007 |
| Diamond | 0.136 | 0.071 | 0.191 | 0.255 | 0.171 | 0.095 | 0.136 | 0.081 | 0.079 | 0.125 | 0.022 | 0.027 |





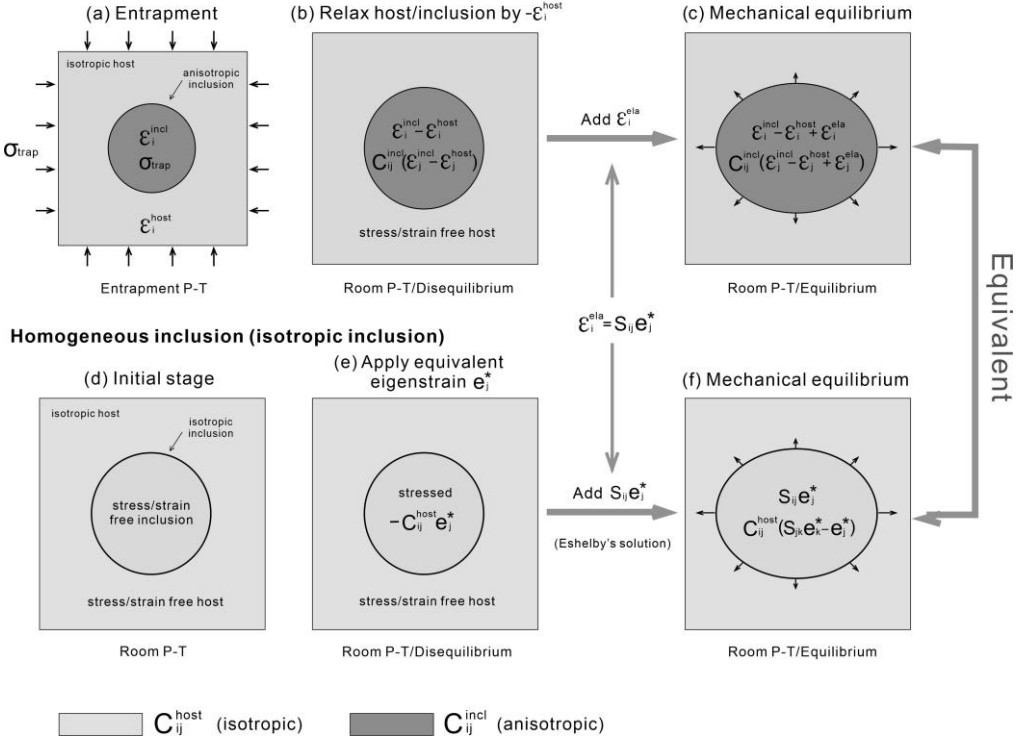

**Fig. 1**. Schematic diagram showing how to obtain the residual stress/strain of anisotropic inclusion in isotropic host. (a) Inclusion-host at entrapment conditions. The stress is homogeneous as $\sigma_{\text{trap}}$ but strains are different as $\varepsilon_i^{\text{incl}}$ and $\varepsilon_i^{\text{host}}$. (b) First, relax the inclusion and host by $-\varepsilon_i^{\text{host}}$ to room *P-T* conditions. Without elastic interaction, the inclusion has strain $\varepsilon_i^{\text{incl}} - \varepsilon_i^{\text{host}}$ and stress $C_{ij}^{\text{incl}}(\varepsilon_j^{\text{incl}} - \varepsilon_j^{\text{host}})$ so the system is not in mechanical equilibrium. (c) Elastic interaction occurs to reach equilibrium by adding strain $\varepsilon_i$ to the inclusion (host also deforms). The residual inclusion stress is $C_{ij}^{\text{incl}}(\varepsilon_j^{\text{incl}} - \varepsilon_j^{\text{host}} + \varepsilon_j)$. (d) Equivalent scenario where the inclusion and host are initially stress free at room *P-T* and they both have isotropic elasticity of $C_{ij}^{\text{host}}$. (e) Equivalent eigenstrains $e_i^*$ are loaded to the inclusion. Without elastic interaction, the inclusion has stress $-C_{ij}^{\text{host}}e_j^*$. Eshelby's method is applied to obtain the final strain state in isotropic inclusion as $S_{ij}e_j^*$ and stress as $C_{ij}^{\text{host}}(S_{jk}e_k^* - e_j^*)$, where $S_{ij}$ is the Eshelby's tensor (Eshelby, 1957). Equivalent eigenstrain method states that by properly choosing $e_i^*$, the relation $\varepsilon_i = S_{ij}e_j^*$ can be satisfied (Mura, 1987, chapter 4). The stress of isotropic inclusion (f) as $C_{ij}^{\text{host}}(S_{jk}e_k^* - e_j^*)$ equals the stress of the anisotropic inclusion (c) as $C_{ij}^{\text{incl}}(\varepsilon_j^{\text{incl}} - \varepsilon_j^{\text{host}} + S_{jk}e_k^*)$ (see Eq. 7). By solving for $e_j^*$, we obtain the residual stress and strain of anisotropic inclusion in isotropic host in Eq. 8.




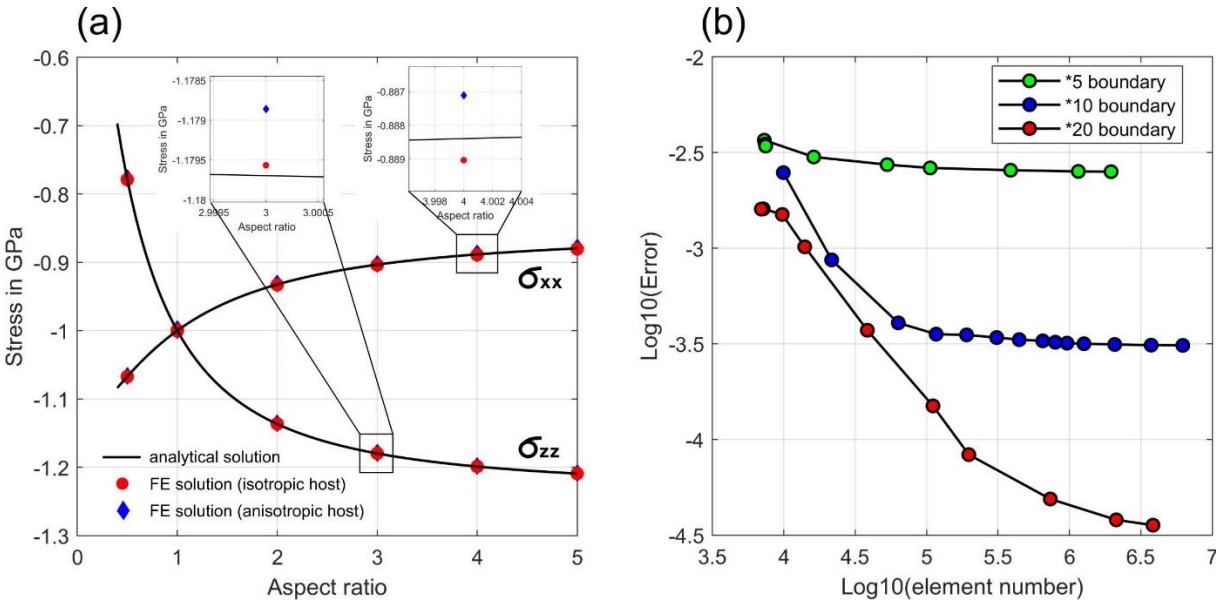

**Fig. 2**. Cross validation results between finite element method and the presented analytical method. (a) Direct comparison of residual stress components calculated with FE method and analytical method as a function of the aspect ratio of a spheroidal inclusion. (b) The normalized unsigned difference of the stress between FE method and analytical method as a function of mesh element number and model domain size. Spherical inclusion is used and boundary distance is set to *5, *10 and *20 the inclusion diameter.






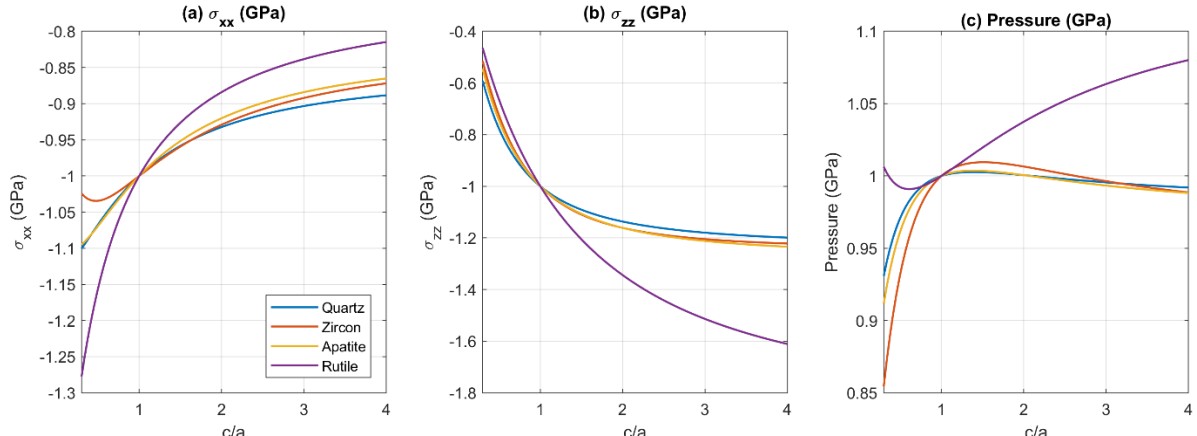

**Fig. 3**. Effect of geometrical aspect ratio of spheroidal inclusion along *c* and *a*-axes on residual stress (*c/a*). (a-b) Stresses $\sigma_{xx}^{res}$ and $\sigma_{zz}^{res}$ as a function of the geometrical *c/a* ratio for quartz, zircon, apatite and rutile inclusions. To isolate the effects of aspect ratio, the eigenstrain are set to produce $\sigma_{xx}^{res} = \sigma_{zz}^{res} = -1$ GPa for the reference spherical inclusion. Any deviation from -1 GPa is due to shape changes. (c) Pressure as a function of the *c/a* ratio. Here, the crystallographic *c*-axes is aligned parallel to the long axis for prolate inclusions and short axis for oblate inclusions. The quartz elastic stiffness tensor is from Heyliger et al., (2003); zircon and diamond from Bass, (1995); rutile from Wachtman et al. (1962); apatite from Sha et al. (1994). Isotropic stiffness tensor of almandine host is from Milani et al. (2015).





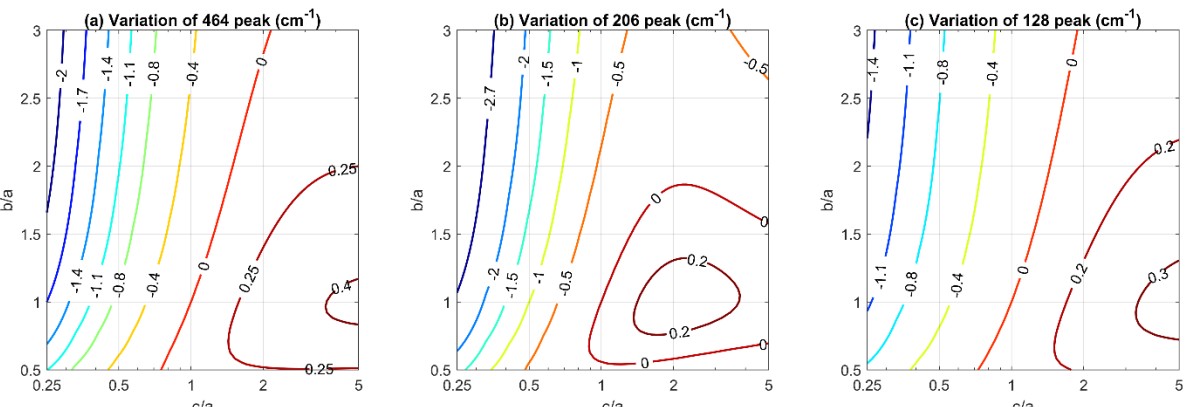

**Fig. 4**. The effect of geometrical aspect ratio of ellipsoidal quartz inclusion for $c/a$-axes and $b/a$-axes on Raman wavenumber shift entrapped in garnet host. The contours show the variation of wavenumber shift compared to perfectly spherical quartz inclusion ($c/a=1$, $b/a=1$). The initial residual inclusion pressure is assumed to be hydrostatic 1 GPa for the reference spherical inclusion. The wavenumber shift variation are due to changing the geometrical aspect ratios of $c/a$ and $b/a$ axes. The Raman shifts are calculated using the residual strain and the Grüneisen tensor in Murri et al. (2018). For $c/a>1$, the inclusion is prolate and for $c/a<1$, the inclusion is oblate. The stiffness tensor of quartz at room $P$-$T$ is from Heyliger et al., (2003) and almandine garnet from Milani et al. (2015).





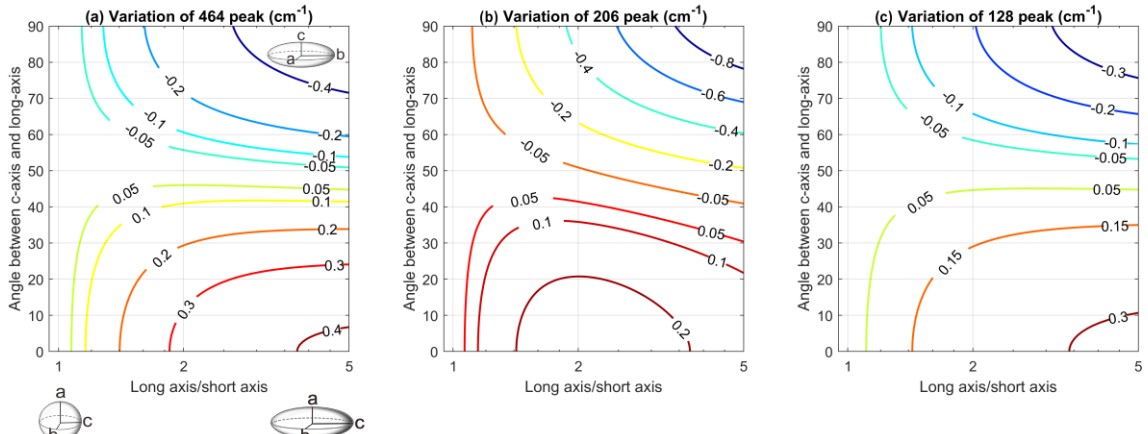

**Fig. 5**. The effect of varying the crystallographic orientation (*c*-axis) with respect to the geometrical long axis of a prolate spheroidal quartz inclusion. The contours show the variation of wavenumber shift compared to perfectly spherical quartz inclusion *c/a*=1 (in this case the crystallographic orientation does not matter). The horizontal axis represents the aspect ratio of the spheroidal inclusion, and the vertical axis shows the angle between the crystallographic *c*-axis and the geometrical long axis. In the plot, *c*-axis is allowed to shift from parallel to the geometrical long axis to parallel to geometrical short axis of the spheroidal inclusion. The driving eigenstrain is set to produce a hydrostatic residual pressure of 1GPa in the reference spherical inclusion.





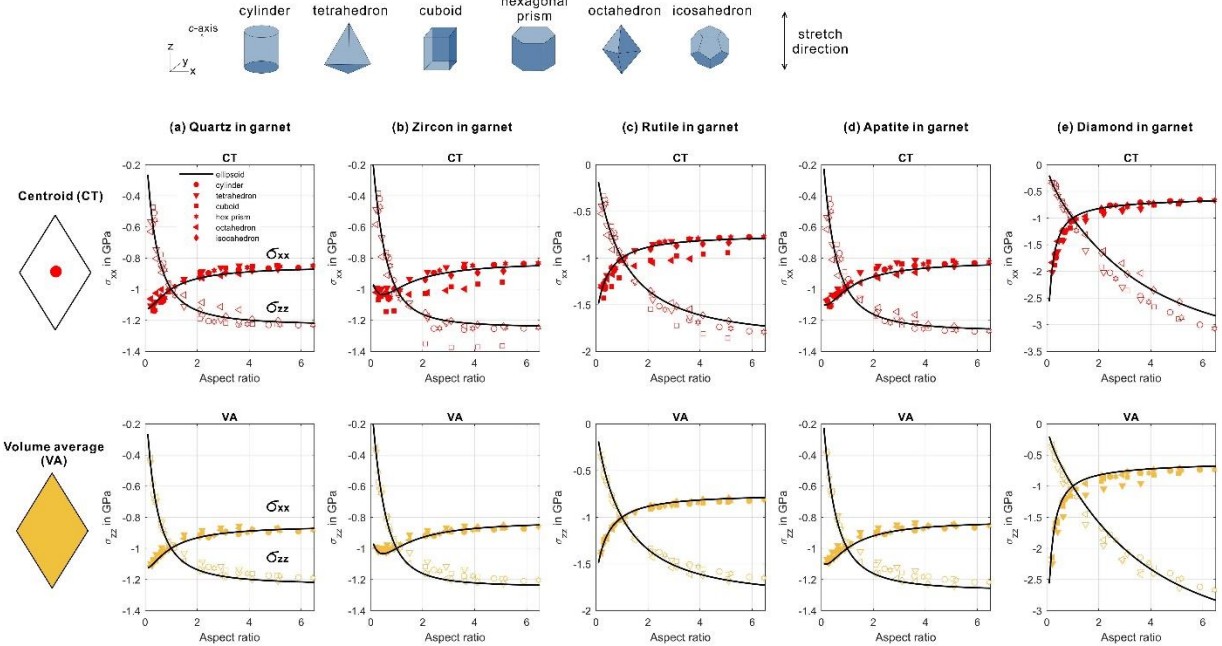

**Fig. 6**. Finite element stress of various inclusion shapes (symbols) compared to the stress of effective spheroidal inclusion based on analytical method (black curves). The effective aspect ratio of inclusion shape is calculated based on the fitting method of Chaudhuri and Samanta, (1991) and Li et al. (1999) (see Appendix). The inclusion is loaded with eigenstrain that generates 1 GPa compressive hydrostatic residual stress for spherical shape. The variation of stress is only caused by the shape change. The *c*-axis coincides with the streching direction. The red dots correspond to the stress at the inclusion's

centroid (CT), the orange dots correspond to the volumetric average (VA) of the entire inclusion. The anisotropic elastic stiffness tensor are listed in the caption of Fig. 3. The root mean square deviation (RMSD) for each inclusion shape and inclusion mineral phase is given in Table 1. The raw FE stress data can be found in supplementary excel file.



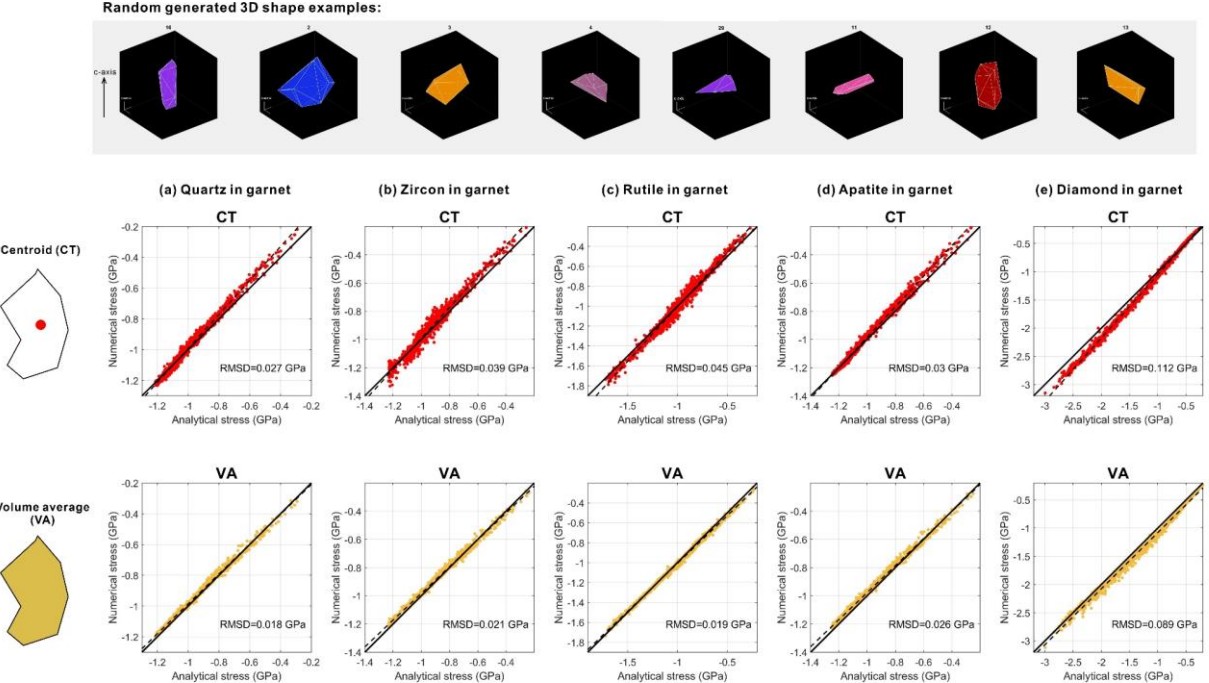

**Fig. 7**. Randomly generated 500 inclusion shapes (top panel for examples) calculated with finite element method (vertical axis) and analytical method (horizontal axis). All three normal stress components are plotted together in each diagram. The crystallographic *c*-axis's orientation is randomly chosen along one of the geometrical principal axes. The red and orange dots show the comparison of FE (numerical) results and analytical results for the normal stress components. Each dot represents a normal stress component calculated for one randomly generated inclusion shape. The red dots show the stress evaluated at the centroid point (CT), the orange dots show the volumetrically averaged (VA) stress within the inclusion. The raw data can be found in supplementary excel file and the generated 3D random shape can be viewed in the *.gif* file.



## Appendix

### Calculate lattice strain at entrapment conditions

When the inclusion and host crystalized at entrapment conditions, they are considered to be stressed and strained by taking
room $P$-$T$ condition as the reference state. Therefore, it is possible to calculate their strain state using lattice parameters $a$, $b$ and $c$ relative to the reference state $a_0$, $b_0$ $c_0$. For cubic, tetragonal and orthorhombic symmetry systems (or hexagonal and trigonal minerals with symmetry being imposed), the lattice strains can be easily expressed following Eq. 1. For triclinic and monoclinic symmetry systems, the basis vectors of unit cell are not all parallel to the Cartesian coordinates $x$, $y$ and $z$. To obtain the lattice strain, we need to transform the coordinate from $a$, $b$ and $c$ into $x$, $y$ and $z$. We follow the method from
Ohashi and Burnham (1973) to calculate the strain components based on the lattice parameters. Here, a short description on the involved equations is given and detailed can be found the appendix of Ohashi and Burnham (1973). This transformation considers the crystallographic $c$-axis parallel to the Cartesian $z$-axis and crystallographic $a$*-axis parallel to the Cartesian $x$-axis.

The matrix $Q_0$ that relates the basis vectors of undeformed crystallographic $a_0$, $b_0$ and $c_0$-axes at reference room $P$-$T$
conditions to Cartesian coordinates is as follows:

$$Q_0 = \begin{bmatrix} \frac{a_0 p_0}{\sin(\alpha_0)} & \frac{a_0(\cos(\gamma_0)-\cos(\alpha_0)\cos(\beta_0))}{\sin(\alpha_0)} & a_0\cos(\beta_0) \\ 0 & b_0\sin(\alpha_0) & b_0\cos(\alpha_0) \\ 0 & 0 & c_0 \end{bmatrix}$$ (A1)

$$p_0 = [1 - \cos^2(\alpha_0) - \cos^2(\beta_0) - \cos^2(\gamma_0) + 2\cos(\alpha_0)\cos(\beta_0)\cos(\gamma_0)]^{1/2}$$

To obtain the similar transformation matrix relating the deformed crystallographic axes at entrapment conditions to Cartesian coordinates can be easily done by replacing $a_0$, $b_0$ $c_0$, $\alpha_0$, $\beta_0$, $\gamma_0$ measured at reference $P$-$T$ state to $a$, $b$, $c$, $\alpha$, $\beta$, $\gamma$ that are measured at entrapment condition from Eq. A1. This transformation matrix is denoted as $Q_1$. We then calculate the displacement gradient tensor $E$:

$$E = Q_0^{-1}Q_1 - I$$ (A2)

where I is the identity matrix. Without considering the antisymmetric rotation tensor, the infinitesimal Lagrangian strain tensor can be expressed as follows:

$$\varepsilon = (E' + E)/2$$ (A3)

A MATLAB code is provided to perform this calculation (*Calculate_Strain.m*). The input values are the reference lattice parameters measured at room $P$-$T$ conditions and the deformed lattice parameters at the entrapment conditions. The outputs are both the infinitesimal and finite Lagrangian strain tensor reported in MATLAB commend window. The results are





numerically the same compared to the available computer programs such as "STRAIN" program that can be found at website:   https://www.cryst.ehu.es/cryst/strain.html,   or   "Win_Strain"   program   at   the   website: http://www.rossangel.com/text_strain.htm.

**Calculate Eshelby's tensor**

The components of Eshelby's tensor $S_{ij}$ are expressed as functions of the inclusion's principal axes length and the Poisson ratio of the isotropic host $\nu$ (Mura, 1987). A MATLAB script is provided to calculate the Eshelby's tensor (see more details in the following sections for supplementary data).

$$S_{11} = \frac{3a_1^2}{8\pi(1-\nu)}I_{11} + \frac{1-2\nu}{8\pi(1-\nu)}I_1$$

$$S_{22} = \frac{3a_2^2}{8\pi(1-\nu)}I_{22} + \frac{1-2\nu}{8\pi(1-\nu)}I_2$$

$$S_{33} = \frac{3a_3^2}{8\pi(1-\nu)}I_{11} + \frac{1-2\nu}{8\pi(1-\nu)}I_3$$

$$S_{12} = \frac{a_2^2}{8\pi(1-\nu)}I_{12} - \frac{1-2\nu}{8\pi(1-\nu)}I_1$$

$$S_{21} = \frac{a_1^2}{8\pi(1-\nu)}I_{12} - \frac{1-2\nu}{8\pi(1-\nu)}I_2$$

$$S_{13} = \frac{a_3^2}{8\pi(1-\nu)}I_{13} - \frac{1-2\nu}{8\pi(1-\nu)}I_1$$

$$S_{31} = \frac{a_1^2}{8\pi(1-\nu)}I_{13} - \frac{1-2\nu}{8\pi(1-\nu)}I_3$$

$$S_{23} = \frac{a_2^2}{8\pi(1-\nu)}I_{23} - \frac{1-2\nu}{8\pi(1-\nu)}I_2$$

$$S_{32} = \frac{a_3^2}{8\pi(1-\nu)}I_{23} - \frac{1-2\nu}{8\pi(1-\nu)}I_3$$

$$S_{44} = \frac{a_2^2-a_3^2}{16\pi(1-\nu)}I_{23} + \frac{1-2\nu}{16\pi(1-\nu)}(I_2 + I_3)$$

$$S_{55} = \frac{a_1^2-a_3^2}{16\pi(1-\nu)}I_{13} + \frac{1-2\nu}{16\pi(1-\nu)}(I_1 + I_3)$$

$$S_{66} = \frac{a_1^2-a_2^2}{16\pi(1-\nu)}I_{12} + \frac{1-2\nu}{16\pi(1-\nu)}(I_1 + I_2)$$

(A4)





where $a_1$, $a_2$ and $a_3$ are the lengths of three principal axes of the inclusions and they follow the order of $a_1 > a_2 > a_3$. In case this order needs to be changes, a simple 90 degree rotation can be executed on $S_{ij}$. The provided code in supplementary data automatically perform such rotation to adjust the axes into correct order. The required tensors $I_i$ and $I_{ij}$ are evaluated as follows:

$$I_1 = \frac{4\pi a_1 a_2 a_3}{(a_1^2 - a_2^2)(a_1^2 - a_3^2)^{1/2}} [F(\theta, k) - E(\theta, k)]$$

$$I_3 = \frac{4\pi a_1 a_2 a_3}{(a_2^2 - a_3^2)(a_1^2 - a_3^2)^{1/2}} [\frac{a_2(a_1^2 - a_3^2)^{1/2}}{a_1 a_3} - E(\theta, k)]$$

$$I_2 = 4\pi - I_1 - I_3$$

$$I_{12} = \frac{I_2 - I_1}{a_1^2 - a_2^2}$$

$$I_{13} = \frac{I_3 - I_1}{a_1^2 - a_3^2}$$

$$I_{23} = \frac{I_3 - I_2}{a_2^2 - a_3^2}$$

$$I_{11} = \frac{1}{3}(\frac{4\pi}{a_1^2} - I_{12} - I_{13})$$

$$I_{22} = \frac{1}{3}(\frac{4\pi}{a_2^2} - I_{12} - I_{23})$$

$$I_{33} = \frac{1}{3}(\frac{4\pi}{a_3^2} - I_{13} - I_{23})$$

(A5)

where the functions $F(\theta, k)$ and $E(\theta, k)$ denote the incomplete elliptic integrals of the first and second kind:

$$F(\theta, k) = \int_0^\theta \frac{dw}{\sqrt{1 - k^2 sin^2(w)}}$$

$$E(\theta, k) = \int_0^\theta \sqrt{1 - k^2 sin^2(w)} dw$$

$$\theta = arcsin(\sqrt{1 - \frac{a_3^2}{a_1^2}})$$

$$k = \sqrt{(a_1^2 - a_2^2)/(a_1^2 - a_3^2)}$$

(A6)

The integrals $F(\theta, k)$ and $E(\theta, k)$ are evaluated using the method of the arithmetic-geometric mean (Abramowitz and Stegenm, 1964, chapter 17). Once $F(\theta, k)$ and $E(\theta, k)$ are obtained, $I$ can be computed and substituted into the Eshelby's tensor.





**Fit arbitrary inclusion shape with effective ellipsoid**

The method (details see Chaudhuri and Samanta, 1991; Li et al., 1999) requires a 3D data/image of the inclusion consist of regular cubic voxels, which has volume $\Delta$ in each voxel and coordinate $x$, $y$, $z$ at the center of each voxel. The inclusion is
denoted as domain $R$. Its second-order moment matrix is calculated as follows:

$$I_x = \iiint_R (y^2 + z^2)dxdydz \approx \sum_{i=1}^n (y_i^2 + z_i^2)\Delta$$

$$I_y = \iiint_R (x^2 + z^2)dxdydz \approx \sum_{i=1}^n (x_i^2 + z_i^2)\Delta$$

$$I_z = \iiint_R (x^2 + y^2)dxdydz \approx \sum_{i=1}^n (x_i^2 + y_i^2)\Delta$$

(A7)

$$I_{xy} = \iiint_R (-xy)dxdydz \approx \sum_{i=1}^n (-x_i y_i)\Delta$$

$$I_{yz} = \iiint_R (-yz)dxdydz \approx \sum_{i=1}^n (-y_i z_i)\Delta$$

$$I_{xz} = \iiint_R (-xz)dxdydz \approx \sum_{i=1}^n (-x_i z_i)\Delta$$

where $x_i$ is the $x$ coordinate of the $i$ th voxel that makes the inclusion, $n$ is the total number of the voxels that makes the inclusion domain $R$. A symmetric 3-by-3 matrix is constructed with the above six components and its eigenvalues are denoted as $I_1$, $I_2$ and $I_3$. The length of major, intermediate and minor axes can be calculated straightforwardly as follows, respectively:

$$a = \sqrt{\frac{5}{2I_0}(I_2 + I_3 - I_1)}$$

(A8)

$$b = \sqrt{\frac{5}{2I_0}(I_1 + I_3 - I_2)}$$

$$c = \sqrt{\frac{5}{2I_0}(I_1 + I_2 - I_3)}$$

where $I_0$ is the volume of the shape $R$, which can be straightforwardly calculated as $I_0 = n\Delta$. The orientation of the inclusion's maximal principle axes expressed in direction cosine ($\alpha$, $\beta$ and $\gamma$) is obtained by minimizing the following equation as a function of direction cosine $\alpha$, $\beta$ and $\gamma$ (done in MATLAB with the constrained minimization "*fmincon*" function):

$$F = [\cos(\alpha), \cos(\beta), \cos(\gamma)] \begin{bmatrix} I_x & I_{xy} & I_{xz} \\ I_{xy} & I_y & I_{yz} \\ I_{xz} & I_{yz} & I_z \end{bmatrix} \begin{bmatrix} \cos(\alpha) \\ \cos(\beta) \\ \cos(\gamma) \end{bmatrix}$$

(A9)



where the direction cosine must also satisfy the relation: $\cos(\alpha)^2 + \cos(\beta)^2 + \cos(\gamma)^2 = 1$ (as constraint). We provide the
MATLAB source code (*Fit_Ellipsoid.m*) that performs the fit to any arbitrary shape. The input is a 3D pixelated matrix (*D*)
where 0 is for host and 1 is for inclusion. The matrix *D* describes the shape of an arbitrary inclusion shape. The output is the
best-fitted effective ellipsoid's axes lengths and orientations. As an example, the fit is performed for an ellipsoid and the
result returns the originally prescribed axes lengths.

**Summary of supplementary files**

The supplementary files include codes for calculating: 1) the Eshelby's tensor $S_{ij}$ and the dimensionless matrix $M_{ij}$ (in Eq.
8), which are used to calculate the residual stress or strain (*Calculate_Eshelby.m*); 2) the best-fitted effective ellipsoid's axes
lengths and orientations (*Fit_Ellipsoid.m*). 3) the strain tensor based on lattice parameters at reference room *P-T* conditions
and entrapment conditions for any symmetry systems. Details of the codes are given here for users to apply them. The raw
data from FE simulation are also provided for reproduction of the figures.

**1) Code: *Calculate_Eshelby.m***

The inputs are listed below (italic with underline denotes MATLAB variables):

*C_incl*: the inclusion's anisotropic elastic stiffness tensor at room *P-T*.

*G_host*: the isotropic host's shear modulus at room *P-T*.

*K_host*: the isotropic host's bulk modulus at room *P-T*.

*v_host*: the isotropic host's Poisson ratio calculated from *G_host* and *K_host*.

*a, b, c*: geometrical principal axes lengths of ellipsoidal inclusion. They are parallel to *x*, *y* and *z* coordinate axes. Note they
are not the lattice parameters.

The outputs are as follows:

*S*: the Eshelby's tensor $S_{ij}$

*M*: dimensionless $M_{ij}$ matrix that can be plugged into Eq. 8.

**2) Code: *Fit_Ellipsoid.m***

The inputs are as follows:

*D*: 3D pixelated matrix (3D image) describing the shape of the inclusion. The value 1 is given for pixels within the inclusion
and 0 is given for pixels outside the inclusion.

*dx, dy, dz*: spatial increment of the 3D image, i.e. the size of each 3D cuboidal pixel.





An example is given for ellipsoid fitting. The variables _a, b, c_ are the axes' lengths of the ellipsoid. After running the code, we obtain the best-fitted axes' lengths which are the same as the input.

The outputs of the code are as follows:

_A_: axes' lengths of the best fitted inclusion (sorted from maximal to minimal principle axes of the best-fitted effective ellipsoid)

_B_: directional cosine of the maximal, intermediate and minimal principle axes. Converting directional cosine to angle ($\alpha, \beta$ and $\gamma$) can be easily done by using "_acos_" function in MATLAB.

After running the code, the message "Optimization completed because the objective function is non-decreasing in feasible 690 directions" should be displayed on the command window. This means the run is successful. The best-fitted effective ellipsoid's axes lengths and orientations will also be displayed. As this is a non-linear optimization, local minima may be reach for some special inclusion shape or convergence is not obtained that leads to failure of the run. In this case, the initial guess of the orientation needs to be changed. It may be needed to change "_x0_" variable in the "_Fit_Ellip.m_" function. One may modify the initial guess "_x0_" or contact the author at xinzhong0708@gmail.com for support.

**3) Code: _Calculate_Strain.m_**

This code calculates the lattice strain at entrapment condition using lattice parameters at both reference room conditions and entrapment conditions. The inputs are the lattice parameters at room conditions $a_0$, $b_0$ $c_0$ and $\alpha_0$, $\beta_0$, $\gamma_0$, and lattice parameters at entrapment conditions $a$, $b$, $c$ and $\alpha$, $\beta$, $\gamma$. The output are the strain in Cartesian coordinate system. The Cartesian _x_-axis is parallel to crystallographic _a*_-axis and z-axis parallel to _c_-axis by convention. The reported strain tensors 700 include infinitesimal and finite Lagrangian strain, which are close to each other for small strain problems and the difference is well below the detection limit of any analytical techniques.

**4) Excel raw data for FE model**

The file "_FE_regular.xlsx_" and "_FE_random.xlsx_" are the calculated raw data for section 4.3 and section 4.4. The data can be used to reproduce Fig. 6, Fig. 7 and Table 1.

**5) Animation of random inclusion shape (for section 4.4 "random faceted shape")**

The 3D random inclusion shapes are visualized as cartoon in "_Random_3D_Shape.gif_" file. It can be viewed by dragging into any IE browser. Details of generating the random inclusion shape with random crystallographic orientation is given in the main text.