# Peer review of "Analytical solution for residual stress and strain preserved in anisotropic inclusion entrapped in isotropic host"

_Solid Earth, 2020_

## Referee Comment (RC1) · Anonymous Referee #1 · 5 Nov 2020

Dear editor, I read the manuscript from Zhong and co-authors with great interest. Their manuscript deals with several open questions in the field of elastic thermobarometry and the results provided by the authors can be very useful to the community. Apart from the lengthy, but necessary, theoretical derivations provided by the authors, this manuscript has significant results when it comes to the application of Raman elastic thermobarometry. In addition, I find their new results on the application of the "volumetrically averaged stress" and the "irregularly faceted inclusions" very interesting and of exceptional quality. Finally, it is very rare to find such studies where the authors have tested their analytical work so extensively.

[Figure]

General Comments My major point of criticism concerns the detailed and clear description of the steps involved in the procedure for the calculation of the eigenstrain. I have written in detail my main points below but they concern the clarity of the presentation and not the actual methodology (which is actually based on well established theories). I believe that this part may be difficult to the petrological community and some things which are considered basic in other fields need to be explained in more detail here.

Specific Comments l. 27-28: Please be more specific that you refer to cases with garnet hosts.

l. 36-37: Somewhere here is implied that you need an elastic model to recover the entrapment conditions. The reason I make this distinction is because one may confuse the elastic model that can be done to convert strains (from vibrational mode shifts) to pressure, with the elastic model that is performed to calculate entrapment conditions from residual P. Please be more specific.

l. 40: Zhang's model allows non-infinite host, it is more general.

l. 50: "no numerical software or programming is required", Theoretically, one could do things by hand (even FEM), in addition, plotting the analytical solution may be more efficient by a software such as Matlab. I would rephrase as "..is that the solution is exact and can be obtained rapidly..."

l. 51-52 Please add a relevant reference that relates MC with Uncertainty propagation.

l. 69: "The MATLAB code", which MATLAB code, of the analytical solution, of the FE? Please be more specific.

l. 118-120: I do not quite follow what you mean here ("The thermal effects..."). Could you please develop a bit more?

l. 121: I think that this way of formulating may be confusing. My point is that the eigenstress is what it is (as defined in Eq. 2) and any mineral can have it no matter how stiff or soft. However, if I understood your argument correctly, for a very soft

inclusion in an infinitely rigid host, its eigenstress will be equal to its actual stress. The reason why I would be so specific is because the way its currently expressed it looks that eigenstresses can be defined only for soft minerals in rigid inclusions.

l. 136, what is the "equivalent eigenstrain"? How it is different from the previous one. Could you be more specific? Please also add that the equivalent eigenstrain is yet unknown and needs to be solved for. I think this part deserves a bit more development so clarify some details that may not be obvious to the reader who is not familiar to the Eshelby solution. In that case I would highlight if the stress balance solved for or if it is satisfied by the solution (i.e. is given). In addition, I would highlight that eigenstrain is needed in order to have "equivalent" loading conditions.

l. 143, as before: please mention how the Eshelby tensor is obtained in general, i.e. it is solved for, is it known a-priori (e.g. from Mura, 1987)

l. 171, please add "phonon-mode" in the Gruneisen tensor so that this is not confused with the macroscopic definition.

l. 173, please add "(pressure)" after stress, since you are using it later.

l. 238, 256, like in l. 171

l. 283, which "size" you are referring to? The largest? A mean size?

l. 370, thus the results using rutile should be viewed with caution since they potentially have large errors.

Minor things l. 45: "long time" is relative in geology. I would be more specific, i.e. for more than 50 years

l. 139-141, I would break this sentence in smaller parts.

l. 150, I would suggested reformatting, "equals" -> ".. to be equal to.."

l. 155, I suppose that this is actually a system of equations that gives you all the

eigenstrain components. I would add some brackets here to emphasize this point.

l. 204, I would rather replace "space" with "medium"

l. 290, Please give the formula of Root Mean Square in the text or in Appendix

l. 560, "is aligned" -> "are aligned"

l. 567, please add in brackets the garnet composition (e.g. alm)

l. 570 as in l. 171

---

## Short Comment (SC1) · 16 Nov 2020

Comments to Zhang et al., "Analytical solution for residual stress and strain preserved in anisotropic inclusion entrapped in isotropic host" submitted to Solid Earth.

The analysis developed in this manuscript to calculate the Raman shifts developed in ellipsoidal inclusions is correct under the assumption that the symmetry of the inclusion crystal is not broken by the strains imposed by the host crystal. Under these circumstances, it agrees with the extensive analyses published in both the materials science and geological literature.

[Figure]

With an isotropic host crystal (and we agree that normal metamorphic silicate garnets can be treated as being elastically isotropic for these purposes) the Eshelby solution for inclusion stress and strain shows that the symmetry of an ellipsoidal inclusion crystal will be broken when the crystallographic axes of the inclusion crystal do not coincide with the principal axes of its ellipsoidal shape. The manuscript is also correct in stating that the symmetry of faceted inclusion crystals will be also be broken; this is the consequence of stress and strain concentration at corners and edges of the inclusion, as well as of the orientation of the crystallographic axes of the inclusion with respect to the shape.

Such symmetry-breaking of the inclusion crystal must also affect the values of the components of the phonon-mode Grüneisen tensor which determine its Raman shifts arising from the strains applied to the inclusions. These Grüneisen tensors have only been determined for a limited number of crystals. Recent DFT calculations of these Grünesien tensors (Murri et al. 2018 for quartz; Stangarone et al. 2019 for zircon, Musiyachenko et al. 2020 for rutile) explicitly assume that the symmetry of the crystal is preserved. That means that the unit-cell strains are constrained as e1 = e2 to preserve the equivalence of the a- and b-axes of these uniaxial crystals. Therefore, these tensors cannot be applied to strain states where the e1 component is different from e2, or those with non-zero shear components, which will arise from the mechanical states presented in Figures 4 and 5 of this manuscript.

The magnitude of the effect of symmetry-breaking on the Grüneisen tensors of minerals has not been calculated in recent DFT simulations. But there is direct experimental evidence that it can be significant compared to the shifts without symmetry breaking (e.g. Briggs and Ramdas, 1977, on quartz). If symmetry-breaking was not an issue, then the Raman peaks of cubic host minerals such as diamond and garnets would not exhibit any change in the deviatoric strain fields around inclusions (e.g. Angel et al 2019). But the Raman shifts in diamond (e.g. Nasdala et al., 2005) and garnets (Campomenosi et al., 2020) around inclusions have been measured and are significant. They are correlated with the symmetry-breaking visible in thin sections as optical birefringence haloes. Therefore, the calculations in the current manuscript of Raman shifts of inclusions whose symmetry is broken is not correct. The magnitude of the error is unknown.

In summary, for the cases in which the symmetry of the inclusion crystal is not broken, this manuscript provides results that are in agreement with previous studies with a variety of methods. For inclusions whose symmetry is broken, this manuscript does not allow for the additional Raman shifts that will arise from the symmetry breaking. This means that Figures such as 4 and 5 should not be used to interpret the Raman shifts in quartz inclusions trapped in garnets and the authors should clearly identify in the manuscript all of their examples and calculations in which the inclusion symmetry is broken.

Matteo Alvaro, Pavia; Ross J. Angel, Padova; Mattia Mazzucchelli, Mainz.

References

Angel RJ, Murri M, Mihailova B, Alvaro M (2019) Stress, strain and Raman shifts. Zeitschrift für Kristallographie, 234, 129-140.

Briggs RJ, Ramdas AK (1977) Piezospectroscopy of the Raman spectrum of a-quartz. Physical Review B, 16, 3815-3826.

Campomenosi N, Mazzucchelli ML, Mihailova B, Angel RJ, Alvaro M (2020) Using polarized Raman spectroscopy to study the stress gradient in mineral systems with anomalous birefringence. Contributions to Mineralogy and Petrology, 175:16.

Murri, M., Mazzucchelli, M.L., Campomenosi, N., Korsakov, A. V., Prencipe, M., Mihailova, B.D., Scambelluri, M., John, A.R., Matteo, A., Angel, R.J., Alvaro, M., (2018) Raman elastic geobarometry for anisotropic mineral inclusions. Am. Mineral. 103, 1869–1872.

Musiyachenko KA, Murri M, Prencipe M, Angel RJ, Alvaro M (2020) A new Grüneisen

tensor for rutile and its application to host-inclusion systems. American Mineralogist, submitted.

Nasdala, L., Hofmeister, W., Harris, J.W., and Glinnemann, J. (2005) Growth zoning and strain patterns inside diamond crystals as revealed by Raman maps. American Mineralogist, 90, 745–748.

Stangarone C, Alvaro M, Angel R, Prencipe M, Mihailova BD (2019) Determination of the phonon-mode Grüneisen tensors of zircon by DFT simulations. European Journal of Mineralogy, 31, 685-694.

---

## Referee Comment (RC2) · Anonymous Referee #2 · 30 Nov 2020

The manuscript of Zhong et al. presents analytical and numerical solutions for the deformation and stress of ellipsoidal inclusions in an infinite host. These solutions are applied to so-called Raman elastic thermobarometry, which is a method to estimate the peak P-T conditions of exhumed rocks. This thermobarometry is an alternative method with respect to P-T estimates based on thermodynamic Gibbs energy minimizations and is, hence, important to validate and cross-check P-T estimates obtained from different methods. The authors present solutions for an anisotropic inclusion in an isotropic host and further present approximate solutions for so-called faceted inclusions (inclusions with corners). The analytical solutions are tested with numerical simulations based on the finite element method. The Raman elastic thermobarometry is an impor-

tant and more and more applied method to estimate P-T conditions of exhumed rocks and is, hence, of interest for a wide readership. The authors also provide several of their numerical algorithms, which allows readers to reproduce the presented results and to apply these algorithms for their own research. The open access to these algorithms is a great benefit of this contribution. However, the authors should discuss in more detail the limits of applicability of their solutions and potential magnitudes of errors when applied to natural host-inclusion studies, which are likely more complex. Ideally, the authors should provide something like a "check-list" for the application of their solution to natural host-inclusion systems. I have also read the comment to this manuscript by Angel et al., which discusses in detail some limits of the presented models, for example arising due to different orientations of the axes of crystallographic orientations and the principal axis of the ellipsoidal shape. I find this comment very useful and urge the authors to clearly explain and discuss these limitations. Making algorithms available is great for the research community, but always generates the risk that users may apply such algorithms wrongly to natural systems for which the algorithms are actually not correctly applicable. Therefore, the authors should address the limitations and applicability of their solutions in detail during a revision of their manuscript. Apart from this major comment, I have a few minor comments, which the authors might also consider during a revision.

Minor comments:

Line 86-88: These are strong assumptions for the stage of entrapment. Maybe these assumptions could be discussed and justified in the Discussion section.

Line 97: For readers not expert in anisotropy in minerals, it would be useful to explain the angles, maybe even with a little sketch showing the anisotropy axes and the corresponding angles.

Line 107: Please explain what is the PVT relationship. Best would be to just add the formula to avoid any ambiguity.

Line 171: Could you add a sentence explaining the origin of the Grüneisen tensor for the non-specialists. For example, is this tensor derived from theoretical calculations or determined from experiments?

Line 214: A main result is quantifying the impact of the aspect ratio. However, the impact of the aspect ratio is not very transparent from the presented equations. Is there a possibility to provide an equation, which shows the impact of the aspect ratio on the Eshelby tensor clearer, or in a more transparent way?

Line 235-231: Could you provide a simple and/or intuitive explanation why the aspect ratio is least sensitive for quartz but most sensitive for rutile. What is the controlling mechanical difference between quartz and rutile responsible for the different sensitivity?

Line 242: Could you provide a typical value of a wavenumber variation, which "defines" the transition from significant to insignificant variation? Maybe as percentage with respect to the corresponding Raman peak.

Line 272: Please add a sentence explaining what is the second-order moment and why the second-order moment is needed and not the first-order moment.

Line 305: So I guess "interestingly" implies that you did not expect such better approximation. Could you provide now an explanation why you got this better approximation, or do you still not know why this approximation is better?

Line 389: comma instead of point.

Conclusions: The conclusion section could be shortened by stating only the main conclusions and the main new results.

Numerical codes: The Matlab script "Fit_Ellipsoid" uses, for example, the command "syms" which requires the Symbolic Math Toolbox; so this script cannot be run with a basic Matlab license. It would be great if the authors could modify the codes, if possible, so that they can be used also with a basic Matlab student license.

---

## Author Comment (AC1) · 8 Jan 2021

We thank the reviewer for the very positive and helpful comments. Below, we provide point-by-point replies to the comments. Our replies are given in blue and the original comments from the reviewers are in black.

Dear editor, I read the manuscript from Zhong and co-authors with great interest. Their manuscript deals with several open questions in the field of elastic thermobarometry and the results provided by the authors can be very useful to the community. Apart from the lengthy, but necessary, theoretical derivations provided by the authors, this manuscript has significant results when it comes to the application of Raman elastic

thermobarometry. In addition, I find their new results on the application of the "volumetrically averaged stress" and the "irregularly faceted inclusions" very interesting and of exceptional quality. Finally, it is very rare to find such studies where the authors have tested their analytical work so extensively.

General Comments

My major point of criticism concerns the detailed and clear description of the steps involved in the procedure for the calculation of the eigenstrain. I have written in detail my main points below but they concern the clarity of the presentation and not the actual methodology (which is actually based on well-established theories). I believe that this part may be difficult to the petrological community and some things which are considered basic in other fields need to be explained in more detail here.

Specific Comments

l. 27-28: Please be more specific that you refer to cases with garnet hosts.

Revised as suggested.

l. 36-37: Somewhere here is implied that you need an elastic model to recover the entrapment conditions. The reason I make this distinction is because one may confuse the elastic model that can be done to convert strains (from vibrational mode shifts) to pressure, with the elastic model that is performed to calculate entrapment conditions from residual P. Please be more specific.

Revised as suggested.

l. 40: Zhang's model allows non-infinite host, it is more general.

Corrected. We removed the word "infinite". The existing model considers an isotropic inclusion in isotropic host.

l. 50: "no numerical software or programming is required". Theoretically, one could do things by hand (even FEM), in addition, plotting the analytical solution may be more

efficient by a software such as Matlab. I would rephrase as "..is that the solution is exact and can be obtained rapidly..."

Revised as suggested.

l. 51-52 Please add a relevant reference that relates MC with Uncertainty propagation.

Reference about MC has been added.

l. 69: "The MATLAB code", which MATLAB code, of the analytical solution, of the FE? Please be more specific. We have revised this sentence to clarify the point. We only present the MATLAB codes for calculating the 1) Eshelby tensor, 2) lattice strain and 3) effective ellipsoidal shape. No FE code is given. This is made clear now. l. 118-120: I do not quite follow what you mean here ("The thermal effects..."). Could you please develop a bit more?

We have revised this sentence. We realized that thermal effect can be confusing as it is often related to volume change, but we are here referring to the T dependence of the stiffness tensor, which has no effect on the final result. This is made clear here now.

l. 121: I think that this way of formulating may be confusing. My point is that the eigenstress is what it is (as defined in Eq. 2) and any mineral can have it no matter how stiff or soft. However, if I understood your argument correctly, for a very soft inclusion in an infinitely rigid host, its eigenstress will be equal to its actual stress. The reason why I would be so specific is because the way it's currently expressed it looks that eigenstresses can be defined only for soft minerals in rigid inclusions.

We see the confusion here. We have revised this sentence. The eigenstress can be understood as the equilibrated stress for an infinitely soft inclusion in an infinitely stiff host. But for practical inclusion and host system, it is just an internal stress that loads the system.

l. 136, what is the "equivalent eigenstrain"? How it is different from the previous one. Could you be more specific? Please also add that the equivalent eigenstrain is yet

unknown and needs to be solved for. I think this part deserves a bit more development so clarify some details that may not be obvious to the reader who is not familiar to the Eshelby solution. In that case I would highlight if the stress balance solved for or if it is satisfied by the solution (i.e. is given). In addition, I would highlight that eigenstrain is needed in order to have "equivalent" loading conditions.

We have added several new sentences here to clarify the concept of the equivalent eigenstrain and the use of the Eshelby's tensor, which transforms the loaded inclusion eigenstrain into the final strain that is under mechanical equilibrium with the host.

l. 143, as before: please mention how the Eshelby tensor is obtained in general, i.e. it is solved for, is it known a-priori (e.g. from Mura, 1987)

We have added some new sentences here to explain that this tensor is taken as known a-priori based on the previous work of e.g. Mura.

l. 171, please add "phonon-mode" in the Gruneisen tensor so that this is not confused with the macroscopic definition.

Added.

l. 173, please add "(pressure)" after stress, since you are using it later.

Added.

l. 238, 256, like in l. 171

Added. We also checked the rest of the text for similar points.

l. 283, which "size" you are referring to? The largest? A mean size?

Revised. We refer to the boundary of the model and the radius of the inclusion.

l. 370, thus the results using rutile should be viewed with caution since they potentially have large errors.

Revised as suggested.

Minor things l. 45: "long time" is relative in geology. I would be more specific, i.e. for more than 50 years

Revised as suggested.

l. 139-141, I would break this sentence in smaller parts.

This sentence is broken into two separate shorter sentences.

l. 150, I would suggested reformatting, "equals" -> ".. to be equal to.."

Revised as suggested.

l. 155, I suppose that this is actually a system of equations that gives you all the eigenstrain components. I would add some brackets here to emphasize this point.

The reviewer is right and we have changed 'equation' into 'system of equations'.

l. 204, I would rather replace "space" with "medium"

Revised as suggested.

l. 290, Please give the formula of Root Mean Square in the text or in Appendix

The definition of RMSD is now given in the bracket at its first appearance in the main text.

l. 560, "is aligned" -> "are aligned"

Corrected.

l. 567, please add in brackets the garnet composition (e.g. alm)

Revised. We just call it almandine garnet.

l. 570 as in l. 171

Done.

---

## Author Comment (AC2) · 8 Jan 2021

We thank the workers for posting this interesting comment on the symmetry breaking issue. Below, we provide our replies to the comments. Our replies are given in blue and the original comments from the commenters are in black.

The analysis developed in this manuscript to calculate the Raman shifts developed in ellipsoidal inclusions is correct under the assumption that the symmetry of the inclusion crystal is not broken by the strains imposed by the host crystal. Under these circumstances, it agrees with the extensive analyses published in both the materials science and geological literature.

[Figure]

With an isotropic host crystal (and we agree that normal metamorphic silicate garnets can be treated as being elastically isotropic for these purposes) the Eshelby solution for inclusion stress and strain shows that the symmetry of an ellipsoidal inclusion crystal will be broken when the crystallographic axes of the inclusion crystal do not coincide with the principal axes of its ellipsoidal shape. The manuscript is also correct in stating that the symmetry of faceted inclusion crystals will be also be broken; this is the consequence of stress and strain concentration at corners and edges of the inclusion, as well as of the orientation of the crystallographic axes of the inclusion with respect to the shape. Such symmetry breaking of the inclusion crystal must also affect the values of the components of the phonon-mode Grüneisen tensor which determine its Raman shifts arising from the strains applied to the inclusions. These Grüneisen tensors have only been determined for a limited number of crystals. Recent DFT calculations of these Grünesien tensors (Murri et al. 2018 for quartz; Stangarone et al. 2019 for zircon, Musiyachenko et al. 2020 for rutile) explicitly assume that the symmetry of the crystal is preserved. That means that the unit-cell strains are constrained as e1 = e2 to preserve the equivalence of the a- and b-axes of these uniaxial crystals. Therefore, these tensors cannot be applied to strain states where the e1 component is different from e2, or those with non-zero shear components, which will arise from the mechanical states presented in Figures 4 and 5 of this manuscript.

The magnitude of the effect of symmetry-breaking on the Grüneisen tensors of minerals has not been calculated in recent DFT simulations. But there is direct experimental evidence that it can be significant compared to the shifts without symmetry breaking (e.g. Briggs and Ramdas, 1977, on quartz). If symmetry-breaking was not an issue, then the Raman peaks of cubic host minerals such as diamond and garnets would not exhibit any change in the deviatoric strain fields around inclusions (e.g. Angel et al. 2019). But the Raman shifts in diamond (e.g. Nasdala et al., 2005) and garnets (Campomenosi et al., 2020) around inclusions have been measured and are significant. They are correlated with the symmetry-breaking visible in thin sections as optical birefringence haloes. Therefore, the calculations in the current manuscript of Raman

shifts of inclusions whose symmetry is broken is not correct. The magnitude of the error is unknown. In summary, for the cases in which the symmetry of the inclusion crystal is not broken, this manuscript provides results that are in agreement with previous studies with a variety of methods. For inclusions whose symmetry is broken, this manuscript does not allow for the additional Raman shifts that will arise from the symmetry breaking. This means that Figures such as 4 and 5 should not be used to interpret the Raman shifts in quartz inclusions trapped in garnets and the authors should clearly identify in the manuscript all of their examples and calculations in which the inclusion symmetry is broken.

We agree with the commenters that deviatoric stresses will generally impose an effect of the physical properties of minerals. The commenters also correctly point out that the DFT calculation performed by Murri et al. (2018) embeds the property of e1=e2, which mimics the D3 symmetry of quartz. We presume that the commenters argue that when our model predicts e1≠e2, the existing parameterization of the Gruneisen tensor by Murri et al. (2018) should not be applied as the assumption taken before is violated.

First, we want to clarify that the stress-induced symmetry breaking and related impact on physical properties have no adverse effect on the proposed analytical solution based framework to studying inclusion-host mechanics. In our view there are two aspects of the problems: 1) the applicability of the analytical solution and the stress field associated with loads due to inclusion eigenstrain (the main target of the work) and 2) the impact of potential symmetry breaking on the calculated Raman shift. The commenters are mainly concerned about the second part. However, the main outcome of our work, which is the proposed analytical solution framework, is largely unaffected by the stress-induced symmetry-breaking effects. Regarding the symmetry breaking issue, it is not only the Gruneisen tensor that is affected by the symmetry breaking effects due to the presence of general (deviatoric) stress states, when for example shear stress components along the principal crystallographic axes do not necessarily vanish. Any physical property such as elastic stiffness, viscosity and thermal expansivity

etc. is affected as well, and, in general, resulting in substantial challenges as experimental measurements under different level of deviatoric stress need to be performed and fit all the physical properties as a function of individual stress components. To our knowledge, there is no such data related to e.g. quartz in garnet system. As we are not specialists in experiments, there may be technical difficulties associated with such experiments, which the commenters may know better. However, the proposed analytical approach is capable of incorporating an arbitrary stiffness tensor for the inclusion phase, and in particular the cases of oblique crystallographic orientations with respect to the principal geometric axes of the inclusion. The model, in its incremental form, can also embed stress-induced changes to the stiffness tensor, including the effect due to symmetry-breaking (please see below). In fact, the non-linearity of stiffness tensor (both inclusion and host) due to deviatoric stress has not been addressed in any of the available mechanical models relevant to Raman elastic thermobarometry. The analytical framework described in this work has a capacity to deal with this problem for the inclusion phase (however admittedly not for the host phase), and experimental parametrizations of the non-linear dependence of the mineral stiffness tensor components on general (deviatoric) stress states would be greatly appreciated in this context.

The key question is how large the effects due to symmetry-breaking could be in minerals. The commenters referenced the paper of Briggs and Ramdas (1977), where they applied uniaxial stress on single quartz crystal along x, y and z direction to fit the experimental deformation potential (same as Gruneisen tensor components). When the uniaxial force F is along the x or y direction, the symmetry is reduced from D3 to C2 (or C1 if F has arbitrary orientation). In this case, the commenters argue that the Gruneisen tensor should significantly vary. Below, we summarize the experimentally (uniaxial stress) calibrated Gruneisen components with symmetry broken (e1$\neq$e3) and the HF/DFT ab-initio results from Murri et al. (2019). For comparison purpose, we present the ratio between the Gruneisen components along the crystallographic a and c directions. (EXP represents the experimental value from Briggs and Ramdas 1977

| Raman bands | a/c (EXP) | a/c (DFT) |
|---|---|---|
| 464 (A1) | 0.542 | 0.504 |
| 206 (A1) | 0.889 | 0.693 |
| 360 (A1) | -0.550 | -0.689 |
| 1080 (A1) | 0 | 0.06 |

and DFT represents the ab-initio results from Murri et al. 2019). Only A1 vibration mode is reported here.

Although the exact ratio is different, they are still quite comparable and we need to also note that one set is experimental and the other is ab-initio, which is already in good agreement. The experimental data is obtained under uniaxial stress with symmetry broken while the DFT is based on symmetry preserving condition. It is our speculation that the effect of deviatoric stress may not be significant to the Raman shift (also the second-order effect of shear modulus as a function of deviatoric stress). It is also noted that quartz in garnet system has been successfully in numerous literatures cited in the manuscript, showing that the deviatoric stress effect is not so significant at least for quartz case.

We agree that by breaking the mineral symmetry due to applied deviatoric stress, all physical properties, including the Gruneisen and stiffness tensors, may be affected but the amount of the effect is unknown yet due to the lack of experimental data. We would argue that the symmetry breaking effects can be viewed as a subset of a wider class of effects due to stress-induced (non-linear) changes of physical properties. We agree that these effects should be thoroughly studied and the resulting new parameterization could be directly used in improved mechanical models such as the one proposed in this study.

We have added a new paragraph in the manuscript to speculate about the effect of

symmetry breaking. We also would like to point out that we are primarily focused how large the departure of Raman shift is from a spherical case by assuming that the phonon coefficients in front of the e1 and e2 terms are still the same even if e1 is not equal to e2. This is a first-order estimate that is taken here and clarified in the main text. We would be very happy to apply the new calibrations for symmetry broken situation, if they are available, thus we will keep track of the updates of the commenters' research.

References

Angel RJ, Murri M, Mihailova B, Alvaro M (2019) Stress, strain and Raman shifts.Zeitschrift für Kristallographie, 234, 129-140.

Briggs RJ, Ramdas AK (1977) Piezospectroscopy of the Raman spectrum of a-quartz.Physical Review B, 16, 3815-3826.

Campomenosi N, Mazzucchelli ML, Mihailova B, Angel RJ, Alvaro M (2020) Using-polarized Raman spectroscopy to study the stress gradient in mineral systems with-anomalous birefringence. Contributions to Mineralogy and Petrology, 175:16.

Murri, M., Mazzucchelli, M.L., Campomenosi, N., Korsakov, A. V., Prencipe, M., Mihailova, B.D., Scambelluri, M., John, A.R., Matteo, A., Angel, R.J., Alvaro, M., (2018) Raman elastic geobarometry for anisotropic mineral inclusions. Am. Mineral. 103, 1869–1872.

Musiyachenko KA, Murri M, Prencipe M, Angel RJ, Alvaro M (2020) A new Grüneisen tensor for rutile and its application to host-inclusion systems. American Mineralogist, submitted.

Nasdala, L., Hofmeister, W., Harris, J.W., and Glinnemann, J. (2005) Growth zoningand strain patterns inside diamond crystals as revealed by Raman maps. American Mineralogist, 90, 745–748.

Stangarone C, Alvaro M, Angel R, Prencipe M, Mihailova BD (2019) Determination ofthe phonon-mode Grüneisen tensors of zircon by DFT simulations. European Journalof Mineralogy, 31, 685-694

---

## Author Comment (AC3) · 8 Jan 2021

We thank the reviewer for the comments that greatly improve the quality of the work. Our replies to the comments are in blue and the original comments are in black.

The manuscript of Zhong et al. presents analytical and numerical solutions for the deformation and stress of ellipsoidal inclusions in an infinite host. These solutions are applied to so-called Raman elastic thermobarometry, which is a method to estimate the peak P-T conditions of exhumed rocks. This thermobarometry is an alternative method with respect to P-T estimates based on thermodynamic Gibbs energy minimizations and is, hence, important to validate and cross-check P-T estimates obtained

from different methods. The authors present solutions for an anisotropic inclusion in anisotropic host and further present approximate solutions for so-called faceted inclusions (inclusions with corners). The analytical solutions are tested with numerical simulations based on the finite element method. The Raman elastic thermobarometry is an important and more and more applied method to estimate P-T conditions of exhumed rocks and is, hence, of interest for a wide readership. The authors also provide several of their numerical algorithms, which allows readers to reproduce the presented results and to apply these algorithms for their own research. The open access to these algorithms is a great benefit of this contribution.

However, the authors should discuss in more detail the limits of applicability of their solutions and potential magnitudes of errors when applied to natural host-inclusion studies, which are likely more complex. Ideally, the authors should provide something like a "check-list" for the application of their solution to natural host-inclusion systems.

We have added a new section "5. Limitation of applicability" to discuss this issue as suggested by the reviewer. This includes the discussion and limitation of some of the assumptions that we have taken so far, e.g. infinite and isotropic host, linear-elasticity and inclusion shape. This is practically a check list to remind readers of the potential issues of elastic thermobarometry, and we believe there will be certainly more refinement to work on in the future.

I have also read the comment to this manuscript by Angel et al., which discusses in detail some limits of the presented models, for example arising due to different orientations of the axes of crystallographic orientations and the principal axis of the ellipsoidal shape. I find this comment very useful and urge the authors to clearly explain and discuss these limitations.

We have provided a detailed reply to Angel et al. and added new text into the manuscript to address this issue. We agree that this is worth mentioning but, in our view, the symmetry breaking issue has no adverse effect on our work: 1) Its estimated effect on the final Raman shift seems minor after comparing the experimental and DFT calculations (see table above in the reply to Angel et al.). 2) Obtaining the Raman shift using residual strain or stress is intrinsically a post-processing procedure that does not affect our analytical solution at all (so our main focus is not impacted). 3) Lastly, there are simply no available parameterizations of the physical properties such as elastic stiffness, thermal expansivity and particularly the Gruneisen tensor under non-symmetric deviatoric stress.

Making algorithms available is great for the research community, but always generates the risk that users may apply such algorithms wrongly to natural systems for which the algorithms are actually not correctly applicable. Therefore, the authors should address the limitations and applicability of their solutions in detail during a revision of their manuscript. Apart from this major comment, I have a few minor comments, which the authors might also consider during a revision.

Minor comments:

Line 86-88: These are strong assumptions for the stage of entrapment. Maybe these assumptions could be discussed and justified in the Discussion section.

We agree with the reviewer that we use an assumption that upon entrapment, the inclusion and host were subject to the same stress field. However, it can be argued that this assumption is a reasonable one considering that when the inclusion was engulfed by the host during its growth, they must be possess the same stress state under mechanical equilibrium as elastic stress equilibration is a fast process compared to mineral growth. This is made clear in the main text after this sentence.

Line 97: For readers not expert in anisotropy in minerals, it would be useful to explain the angles, maybe even with a little sketch showing the anisotropy axes and the corresponding angles.

We have added a sentence explaining the meaning of the angles.

Line 107: Please explain what the PVT relationship is. Best would be to just add the formula to avoid any ambiguity.

We have added a line describing the PVT relationship. It is difficult to add a simple formula to describe the relationship because: 1) there are many distinct PVT relationships, each applied to different minerals; 2) the relationships are highly non-linear and often implicit, so it is not possible to use one simple formula to describe the PVT relationship. Therefore, we have not added any specific PVT formula to avoid potential confusion.

Line 171: Could you add a sentence explaining the origin of the Grüneisen tensor for the non-specialists. For example, is this tensor derived from theoretical calculations or determined from experiments?

Done. We have added new text here to describe this tensor. Both ab-initio and experimental data on the Gruneisen tensor exist and they seem to match quite well, even if they are done at distinct stress condition. See the table in the reply to Angel et al.

Line 214: A main result is quantifying the impact of the aspect ratio. However, the impact of the aspect ratio is not very transparent from the presented equations. Is there a possibility to provide an equation, which shows the impact of the aspect ratio on the Eshelby tensor clearer, or in a more transparent way?

There is unfortunately no explicit form describing the impact of aspect ratio on the Eshelby tensor (not to mention the final expression for the residual stress), e.g. see the work of Mura 1987, who has attempted and the current formula is at its most simplified form as given in the Appendix. Therefore, we did not change this part.

Line 235-231: Could you provide a simple and/or intuitive explanation why the aspect ratio is least sensitive for quartz but most sensitive for rutile. What is the controlling mechanical difference between quartz and rutile responsible for the different sensitivity?
It's not very intuitive why the rutile is more sensitive to aspect ratio as it requires deriving an explicit form of stress variation from spherical case due to shape change, which cannot be easily done. One possibility is that rutile is highly anisotropic, which makes the residual stress more sensitive to the change of aspect ratio. We have added this into the text. However, we also noted that care must be taken for this explanation because we have not tested all minerals and in case of other minerals, readers are encouraged to perform their own calculation.

Line 242: Could you provide a typical value of a wavenumber variation, which "defines" the transition from significant to insignificant variation? Maybe as percentage with respect to the corresponding Raman peak.

We have added a sentence here to clarify this point. The main point is that as long as the variation stays below the detection limit of standard Raman machine after Gaussian fitting of the Raman band position (e.g. 0.2 cm-1), we consider the effect insignificant. This is made clear in the main text now.

Line 272: Please add a sentence explaining what the second-order moment is and why the second-order moment is needed and not the first-order moment.

We have removed the second-order moment to simplify the text for readers. It is practically a method that minimize the mismatch between the irregular inclusion shape with the effective ellipsoid.

Line 305: So I guess "interestingly" implies that you did not expect such better approximation. Could you provide now an explanation why you got this better approximation, or do you still not know why this approximation is better?

This is an observation that volumetric stress average provides a better approximation than the central point. This is interesting that this measure is better considering the fact that when performing Raman measurement on inclusions, we are in fact averaging over the effective volume under the laser. Therefore, it is interesting and, in fact, useful

that the average volume is a better proxy than just a central point. For an explanation, we can so far provide a speculation that it is due to the consideration of the stress variations at the inclusion-host wall (on the inclusion side) that drive the volumetric average closer to the equivalent stress based on the effective ellipsoid. However, it is difficult to prove it because the inclusion shape is arbitrary and faceted so there is no easy analytical description of the stress field.

Line 389: comma instead of point.

Corrected.

Conclusions: The conclusion section could be shortened by stating only the main conclusions and the main new results.

We have slightly shortened the last section. However, this section is more inclined to geological implications to provide a summary for the geologists who might not be interested in the mathematical derivations, but only the geological relevance. Therefore, we still prefer to keep most of the text so that it is easier to follow and readers may hopefully benefit more for their own petrological works.

Numerical codes: The Matlab script "Fit Ellipsoid" uses, for example, the command "syms" which requires the Symbolic Math Toolbox; so this script cannot be run with a basic Matlab license. It would be great if the authors could modify the codes, if possible, so that they can be used also with a basic Matlab student license.

We thank the reviewer for reminding us this issue. We have revised the code so that now it does not need the symbolic toolbox and everyone with basic MATLAB can use it. We now use the function handle, which is available with standard MATLAB version.

---

## Author Response (AR2)

Dear Prof. Gerya

Thank you very much for your editorial handling of our manuscript entitled "Analytical solution for residual stress and strain preserved in anisotropic inclusion entrapped in isotropic host". In the last round of review, the only change was proposed by the second reviewer who reported that the optimization toolbox for MATLAB is needed to run the uploaded code. In the revised version, we have modified the source code so that any free Student MATLAB licence without optimization toolbox can run it. We have also modified the main text's appendix accordingly to comply to the changes.

Therefore, we would like to resubmit the final version of our manuscript to Solid Earth. We sincerely thank you for your support and handling during the review process, and hope that our work may contribute to the petrological community working on the related topics.

Best Regards

Xin Zhong on behalf of the authors.

03/03/2021 in Berlin